# Fine-grained Control of Generative Data Augmentation in IoT Sensing

**Tianshi Wang, Qikai Yang, Ruijie Wang***, **Dachun Sun, Jinyang Li,
Yizhuo Chen, Yigong Hu, Chaoqi Yang, Tomoyoshi Kimura, Denizhan Kara,
Tarek Abdelzaher**

University of Illinois Urbana-Champaign, USA
```
{tianshi3, qikaiy2, ruijiew2, dsun18, jinyang7, yizhuoc,
yigongh2, chaoqiy2, tkimura4, kara4, zaher}@illinois.edu
```

## Abstract

Internet of Things (IoT) sensing models often suffer from overfitting due to data distribution shifts between training dataset and real-world scenarios. To address this, data augmentation techniques have been adopted to enhance model robustness by bolstering the diversity of synthetic samples within a defined vicinity of existing samples. This paper introduces a novel paradigm of data augmentation for IoT sensing signals by adding fine-grained control to generative models. We define a metric space with statistical metrics that capture the essential features of the short-time Fourier transformed (STFT) spectrograms of IoT sensing signals. These metrics serve as strong conditions for a generative model, enabling us to tailor the spectrogram characteristics in the time-frequency domain according to specific application needs. Furthermore, we propose a set of data augmentation techniques within this metric space to create new data samples. Our method is evaluated across various generative models, datasets, and downstream IoT sensing models. The results demonstrate that our approach surpasses the conventional transformation-based data augmentation techniques and prior generative data augmentation models.

## 1 Introduction

IoT sensing applications aim to detect physical world phenomena in specific environments through time-series data captured by sensors, such as inertial, acoustic, and bioelectrical signals. Given the wide variety of real-world conditions, IoT sensing models often face potential domain shifts and unpredictable variations when deployed [23]. Additionally, collecting and labeling IoT sensing data is expensive because sensing signals are harder for annotators to interpret compared to images, videos, and natural language [32, 34, 21, 22]. Consequently, enhancing model robustness across diverse scenarios with minimal collection overhead becomes a key research focus in the IoT field [55, 56].

A spectrum of data augmentation techniques has been proposed to reduce the need for extensive data collection. The objective is to diversify existing datasets while preserving a plausible data distribution. Conventional data augmentation methods for IoT sensing signals generally apply manually-crafted transformations to create different perspectives of the same sample [58, 12]. Typical transformations include operations in the time domain (such as jittering, rescaling, rotation, and cropping) [12, 29, 39, 50] and the frequency domain (such as spectral flipping, warping, and masking) [58, 11, 38]. They are designed to modify the original data to reflect possible real-world variations, guided by domain knowledge in IoT. However, those transformations can be overly simplistic, failing to capture the

---

*Corresponding author. Email: `ruijiew2@illinois.edu`

nuanced low-level features of realistic IoT signal variance. The resultant artifacts may compromise the authenticity of the synthetic samples, thus offering limited benefits for downstream tasks.

Generative data augmentation techniques have recently been introduced to synthesize highly realistic IoT sensing data [9], such as variational autoencoders (VAEs) [24], generative adversarial networks (GANs) [13], and diffusion models [46]. These techniques use inherent labels, such as activity types in human activity recognition (HAR) and vehicle types in vehicle detection tasks as conditions, for generating data. This process creates new samples with diverse characteristics under the same label, thereby enriching the variety of the training dataset. Despite advances in generating high quality signals, most existing works, if not all, fail to incorporate domain knowledge to guide the generation process, often treating generative models as a black-box.

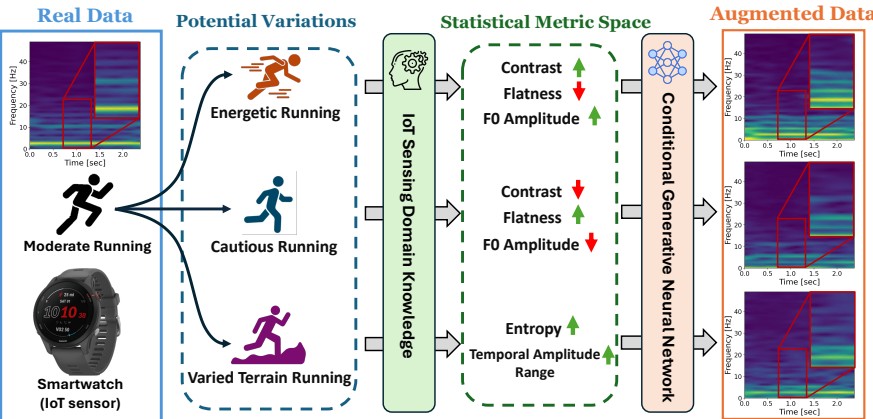

Figure 1: **An Application Example of Our Method.** The insights of the potential variations in the metric space of running signals can guide the generation process of augmented data, thereby enriching the diversity that is missing in the original dataset.

In this paper, we introduce a novel data augmentation paradigm for IoT sensing data that combines domain knowledge control with generative data augmentation methods to advance the synthesis capabilities. We argue that a generative model with fine-grained control over the human-interpretable characteristics of IoT sensing signals can enhance the data augmentation effectiveness on downstream tasks. Our approach begins by leveraging domain expertise in IoT sensing to extract a set of statistical metrics in the time domain, frequency domain, and time-frequency domain, which represent the essential aspects of IoT sensing signals. We call the resultant space determined by those statistical metrics as *metric space*. Subsequently, we develop a suite of data augmentation techniques designed to operate within the metric space.

In Figure 1, we illustrate an application example of our method. Given real data of moderate running in human activity recognition task, we speculate the potential real-world variations of running behaviors and leverage domain knowledge to derive the changes of statistical metrics accordingly. These metrics serve as conditions for a generative neural network to guide the synthetic data generation of other running styles like energetic, cautious and on varied terrain. The generated data is then utilized to augment the original dataset, thereby improving the robustness of downstream IoT sensing model. The strength of our paradigm comes from the precise control over the generation process via the metric space, while ensuring the authenticity of the generated samples through the generative model.

Empirically, we validate the versatility of our augmentation approach with various generative model backbones including a diffusion model and a VAE. We test our approach in three distinct IoT sensing applications: wearable-based human activity recognition, seismic and acoustic-based vehicle detection, and EEG-based harmful brain activity recognition. Our results demonstrate that our method outperforms conventional and generative data augmentations across various downstream models in all the evaluated sensing tasks.

## 2 Related Work

**Traditional Data Augmentation for IoT Sensing.** Many traditional data augmentation methods rely on manually crafted transformations to enhance dataset diversity. As IoT sensing signals typically

involve temporal dynamics while exhibiting periodical patterns, transformations are designed in both time domain [58, 12, 29, 39, 50] and frequency domain [58, 11, 38]. Implementing these augmentation techniques requires specific domain knowledge to determine the variation range that can enhance data diversity without compromising relevance to the intended task. Another data augmentation approach involves similarity interpolation, where new samples are created by interpolating between existing samples, maintaining task invariance in the generated outputs. Data space interpolation techniques, which interpolate directly between raw data samples, have been proposed [5, 63, 37]. Additionally, there has been research on feature space interpolation, which occurs after data has been embedded into latent representations [10, 3, 35]. A primary challenge for this line of work lies in the synthetic data quality. The transformations often represent straightforward, linear variations and may introduce artifacts that reduce the realism of the generated data.

**Generative Data Augmentation for IoT Sensing.** Generative data augmentation utilizes generative models to create varied data [48, 60, 64]. These models, including GANs [49, 31, 6], VAEs [36, 14, 53], and diffusion models [43, 44, 57], generate authentic samples that improve the performance of downstream models and significantly lower the costs associated with collecting real data. However, these prior work treat the generative model as a black box, using it to sample from the learned data distribution with minimal control, typically only conditioned by the inherent labels from the dataset. The lost of control makes the generated results unpredictable, and loses the chance to benefit from domain insights and prior experience. This work is inspired by prior research on conditional generation for human speech and singing signals [25, 4, 16, 33]. In these studies, fine-grained conditions in vocal and/or linguistic features are applied to control acoustic signal generation. However, unlike human speech, IoT sensing time series data carries entirely different semantics and encompasses a broader range of modalities. Condition space interpolation involves the application of interpolation techniques within a condition space, which is constituted by statistical metrics[54]. These augmented metrics are then fed into a generative model to synthesize varied data. Our work significantly advances prior research by redesigning the metric space and enhancing data augmentation techniques within it. We also perform extensive evaluations across diverse generative models and IoT sensing tasks, resulting in a more general data augmentation framework.

# 3 Method

## 3.1 Overview

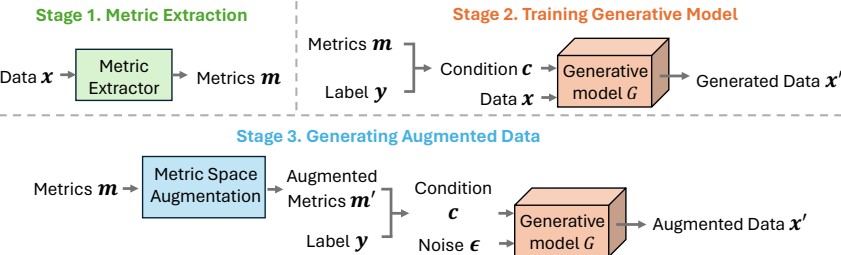

Figure 2: **An overview of the three-stage pipeline.** Given input data $\mathbf{x}$ and label $\mathbf{y}$: Stage 1. Extract statistical metrics $\mathbf{m}$. Stage 2. Train a generative model $G$, conditioned on $\mathbf{m}$ (encapsulated within condition vector $\mathbf{c}$). Stage 3. Generate new data using the augmented metrics $\mathbf{m}'$.

As shown in Figure 2, our fine-grained controllable generative data augmentation has three stages.

In Stage 1, we extract the predefined spectrogram metrics that reflect the critical characteristics of IoT sensing spectrograms. The selection of these metrics is based on domain knowledge in general IoT sensing, capturing essential facets in the IoT signal semantics. These metrics together define the *metric space* that conditions the subsequent generative model.

In Stage 2, the metric values, along with the class label, are embedded into a condition tensor. This tensor then conditions the training of the generative model, guiding it to synthesize spectrograms that adhere to the specified metric values.

In Stage 3, we utilize the fine-grained controllable generative model to augment the original dataset within the metric space. The metrics in this space act as control knobs, allowing for the directional

generation of IoT sensing signals with specific characteristics in the spectrogram semantics. This enables the application of domain knowledge about IoT sensing and downstream tasks to strategically augment the original dataset. We introduce several data augmentation techniques that vary in their reliance on domain knowledge. The augmented conditions (metric values) are input into the trained generative model to produce synthetic samples. These samples, combined with the original dataset, form the augmented dataset used to train the downstream IoT sensing models.

We introduce each stage in details in Section 3.2, 3.3, and 3.4.

## 3.2 Construction of Metric Space

We aim to design a set of statistical metrics which can effectively serve as fine-grained control knobs for data augmentation and easy to be manipulated by the domain experts in IoT sensing.

As IoT sensing signals typically exhibit strong patterns in both time and frequency domain, we convert the original time-series data into 2D spectrograms using short-time Fourier transformation (STFT). We propose metrics as below in three domains: time domain, frequency domain, and time-frequency domain.

**Time domain metric**: Temporal amplitude oscillation commonly exist in IoT sensing signals, which is a strong indicator for the dynamics of the observed physical phenomena. To measure this essential characteristic, we propose using the *temporal amplitude range* in a spectrogram as a metric for controlling the volatility of the generated signal. It straightforwardly informs on the operational limits of the IoT sensing signal, which is essential for a robust generation that must operate within specific amplitude constraints.

**Frequency domain metric**: We introduce *Fundamental frequency (F0) amplitude* as the metric to reflect the frequency domain characteristics of IoT sensing spectrograms. The fundamental frequency is pervasive across various IoT sensing signals. It dictates the dominant periodicity of the signal and is thus essential for capturing the signal's primary oscillatory properties. For example, in a signal captured by the accelerometer of a wearable device on the wrist when the human subject is walking, the fundamental frequency corresponds to the walking cycle.

**Time-frequency domain metrics**: The information distribution across the time-frequency domain is strongly correlated to phenomena such as environmental noises, signal interference, and the harmonic characteristics of the signal source. To effectively capture these aspects, we introduce metrics including *contrast*, *flatness*, and *entropy* that quantify the amplitude distribution and information density within the spectrogram.

*Contrast* quantifies the difference in amplitude between peaks and valleys within the spectrogram. This metric is critical for identifying the dominant signal patterns, allowing for a more nuanced generation of spectrograms where both signal and noise components are realistically balanced.

*Flatness* measures how noise-like a spectrogram is by comparing the geometric mean to the arithmetic mean of the power spectrogram values. A high flatness indicates a uniform distribution of power across the spectrogram, typical of stochastic processes and ambient environmental noises. A low flatness indicates concentration of energy at regions. This results in a spectrogram with interleaving of peaks and valleys, reflecting the presence of tones or harmonics.

*Entropy* measures the randomness of the power distribution within the spectrogram. High entropy suggests a complex, less predictable signal, while low entropy indicates a more structured and tonal signal. This metric is crucial for generating data that mimics the variable information content found in real-world IoT signals, from highly predictable periodic signals to complex, non-periodic disturbances.

The formal definitions of the metrics above can be found in Appendix B.

## 3.3 Conditional Generative Model Training

Given the original dataset, we extract the metric values vector $\mathbf{m}$ from each data sample and embed it with the class label as the condition tensor $\mathbf{c}$. During the training of a conditional generative model $G$, it takes the original data sample $\mathbf{x}$ and the condition tensor $\mathbf{c}$ as inputs. To enforce the alignment of the generated data $\mathbf{x}'$ with the characteristics defined by the given metrics $\mathbf{m}$, a loss term $\mathcal{L}_{metric}$

is taken as a penalty on the difference between the metric values $\hat{\mathbf{m}}$ calculated from $\mathbf{x}'$ and $\mathbf{m}$:

$$\mathcal{L}_{metric} = \sum_{i=1}^{M} w_i(m_i - \hat{m}_i)^2, \tag{1}$$

where $M$ is the total number of metrics, and $w_i$ is a hyperparameter as the weight for a specific metric loss. The total loss $\mathcal{L}$ consists of the original generative model loss $\mathcal{L}_G$ and the metric loss:

$$\mathcal{L} = \mathcal{L}_G + \mathcal{L}_{metric}. \tag{2}$$

In practice, to balance the two loss terms, we empirically adjust the weight $w$ of each metric so that $\mathcal{L}_G : \mathcal{L}_{metric}$ approximately equals to 10:1 at the end of the first training epoch.

### 3.4 Data Augmentation Techniques in Metric Space

To fully leverage the fine-grained controllable generative model to synthesize diverse data samples, effective data augmentation techniques are required. Different from prior approaches, our data augmentation happens in the metric space where the extracted metric values from the original data are manipulated to create variant data. As shown in Figure 3, we propose three data augmentation techniques which require varying levels of IoT domain knowledge, ranging from low to high.

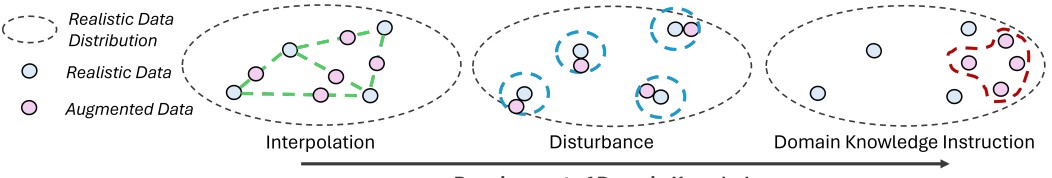

Figure 3: **Data Augmentation Techniques in Metric Space.** In the interpolation method, the green dotted line represents the intermediate value range between two real samples. In the disturbance method, blue circles indicate the range of disturbance. In the domain knowledge instruction method, the red outline denotes specific metric value ranges derived from domain knowledge.

**Interpolation**: A vast amount of prior research has demonstrated that interpolation is an effective data augmentation approach [5, 63, 37, 10, 3]. The intuition is that if the model can correctly classify not only the original samples but also these in-between points, it is likely to perform better on unseen data that fall within the same general manifold. Given a pair of randomly selected samples under the same label $\mathbf{x}_i$ and $\mathbf{x}_j$, we extract their metric values $\mathbf{m}_i$ and $\mathbf{m}_j$. The interpolated metric value $\mathbf{m}_{interp}$ is defined as:

$$\mathbf{m}_{interp} = \beta\mathbf{m}_i + (1 - \beta)\mathbf{m}_j, \tag{3}$$

where $\beta$ is a random number between 0 and 1, independently sampled from a Gaussian distribution for each interpolation. One advantage of interpolation is that it does not require specific domain knowledge to create new data samples.

**Disturbance**: Inheriting from the design of conventional transformation-based data augmentation methods, we disturb the original metric value of a data sample within a predefined varying range $[a, b]$, where $a$ and $b$ are the minimum and maximum percentages of the original metric value. During the metric disturbance, a random number $r$ is sampled from the uniform distribution $U(a, b)$. Given all the metrics as control knobs for creating new samples, users of this augmentation technique can to define the valid disturbance range grounded by domain insights.

**Domain knowledge instruction**: For many IoT sensing applications, deeper insights are available about the downstream model and the potential diversity of the realistic data distribution. In contrast to disturbance, these domain knowledge instructions provide a more precise scope for metric manipulation. This accuracy allows for the synthesis of data that not only enhances the downstream model but also avoids the production of unnecessary variants. To showcase the benefits of incorporating domain knowledge in IoT sensing data augmentation, we present two case studies in Section 4.3.

# 4 Experiments

We evaluate our data augmentation approach on three IoT sensing applications where domain shift issues are prevalent: human-activity recognition, vehicle detection, and harmful brain activity recognition.

## 4.1 Experimental Setup

**Datasets:** (1) **Human-Activity Recognition (HAR)**: We use RealWorld-HAR dataset [47], which aims at classifying 5 human activities based on the accelerometer and gyroscope signals. (2) **Vehicle detection**: We detect 5 vehicle types by deploying a seismic and an acoustic sensor on the ground, monitoring the sound and vibration caused by passing vehicles. This dataset is collected by ourselves [30]. (3) **Harmful brain activity recognition**: We adopt Harmful Brain Activity Classification [18, 19] dataset which comprises electroencephalography (EEG) signals from critically ill patients in hospitals. The objective is to classify 5 types of harmful brain activities.

**Generative Model Backbones**: Technically, our fine-grained controllable generative data augmentation approach is compatible with any conditional generative model. We utilize two types of generative model backbones to validate our approach: (1) *Diffusion Model* [42] (referred to as *Diff* hereafter), and (2) *VAE* [51].

**Baselines:** We compare two categories of data augmentation methods as baselines. ❶ **Traditional data augmentation**: This category includes (1) *data space interpolation* [63], (2) *feature space interpolation* [10], and (3) *time and frequency domain transformation-based method* [38]. ❷ **Label-only conditional generative data augmentation**: We compare with 3 generative models that are only conditioned by the inherent label in the dataset, including (4) *Conditonal Diffusion Model* [42] (cDiff), (5) *Conditional VAE* [51] (cVAE) and (6) *Conditional GAN* [1] (cGAN).

**Downstream Models**: We choose the following two downstream models: (1) *DeepSense* [61] is a deep learning model designed for analyzing IoT signals in the time-frequency domain, featuring multiple convolutional and recurrent layers. (2) *Transformer (Encoder-Only)* [52], which uses self-attention mechanisms to process sequential data and can capture complex dependencies. We only use the encoder component of the original Transformer architecture for classification tasks.

**Data Augmentation Setup**: Each method produce 5 times of synthetic samples comparative the number of the original training samples. For our methods that utilize *disturbance* as the augmentation technique, we set the adjustable range for each metric in between -20% to 20%. This range is reasonably chosen to encompass substantial variance while maintaining invariance critical for the downstream task.

More details about the experimental setup can be found in the Appendix C, D, E, and F.

## 4.2 Overall Performance

Table 1 reports the performance of the 3 variants of our fine-grained controllable generative augmentation comparing with the baselines. *Interp* uses interpolation for augmentation, *Disturb* applies metric disturbance, and *Reconst* generates augmented data by using the actual metric values from the validation set as conditions, showcasing the potential upper bound performance of our method when validation data metrics are perfectly predicted. We highlight the highest-performing cell in the table, excluding the *Reconst* variant.

Across all datasets and two downstream models, our fine-grained controllable generative augmentation consistently outperforms other methods. Typically, the performance improves with the integration of more domain knowledge into the augmentation process. *Interp* operates on the premise that interpolating between real samples enhances dataset diversity, requiring no specific domain knowledge. *Disturb*, on the other hand, involves domain expertise to define the disturbance range, leading to better performance than *Interp*. *Reconst*, which utilizes ground truth metric values from the validation set, achieves the highest performance, illustrating the significant benefits of incorporating domain knowledge about IoT sensing signal characteristics through our designed metrics into the generative augmentation process.

Using the same augmentation technique, the diffusion model generally surpasses the VAE as a backbone. As illustrated in Appendix G.1, VAE generates spectrograms that are less crisp and

Table 1: **General Performance.** Average accuracy and F-1 score of 5 independent runs are reported.

| Augmentation Category | Augmentation Method | Human Activity Recognition | | | | Vehicle Detection | | | | Harmful Brain Activity Recognition | | | |
| --- | --- | --- | --- | --- | --- | --- | --- | --- | --- | --- | --- | --- | --- |
| | | DeepSense | | Transformer | | DeepSense | | Transformer | | DeepSense | | Transformer | |
| | | Acc | F1 | Acc | F1 | Acc | F1 | Acc | F1 | Acc | F1 | Acc | F1 |
| No Augmentation | N/A | 0.7752 | 0.7961 | 0.7835 | 0.8041 | 0.7875 | 0.7768 | 0.7745 | 0.7631 | 0.7001 | 0.6893 | 0.6842 | 0.6633 |
| Fine-grained Controllable Generative Augmentation (ours) | Diff-interp | 0.8125 | 0.8265 | 0.8242 | 0.8338 | 0.8093 | 0.8004 | 0.8005 | 0.7912 | 0.7201 | 0.6945 | 0.7225 | 0.7180 |
| | Diff-disturb | **0.8307** | 0.8333 | 0.8205 | **0.8352** | **0.8200** | **0.8159** | 0.8008 | 0.7944 | **0.7511** | **0.7410** | 0.7335 | 0.7210 |
| | Diff-reconst | 0.8618 | 0.8664 | 0.8591 | 0.8600 | 0.8205 | 0.8094 | 0.8118 | 0.8104 | 0.7588 | 0.7429 | 0.7398 | 0.7220 |
| | VAE-interp | 0.8120 | 0.8123 | 0.8099 | 0.8292 | 0.7836 | 0.7733 | 0.7792 | 0.7653 | 0.6947 | 0.6683 | 0.6991 | 0.6838 |
| | VAE-disturb | 0.8289 | 0.8441 | 0.8151 | 0.8240 | 0.7901 | 0.7785 | 0.7821 | 0.7696 | 0.7016 | 0.6845 | 0.7003 | 0.6801 |
| | VAE-reconst | 0.8372 | 0.8493 | 0.8242 | 0.8377 | 0.8227 | 0.8123 | 0.7897 | 0.7796 | 0.7033 | 0.6837 | 0.7060 | 0.6886 |
| Traditional Data Augmentation | Data Space Interpolation | 0.7894 | 0.7931 | 0.7732 | 0.7759 | 0.7745 | 0.7591 | 0.7410 | 0.7392 | 0.7119 | 0.7034 | 0.7225 | 0.7125 |
| | Feature Space Interpolation | 0.7656 | 0.7708 | 0.7534 | 0.7205 | 0.7730 | 0.7615 | 0.7688 | 0.7510 | 0.6903 | 0.7059 | 0.6821 | 0.6774 |
| | Time and Frequency Domain Transformation | 0.8011 | 0.8117 | 0.8155 | 0.8176 | 0.8024 | 0.7922 | 0.7721 | 0.7662 | 0.7228 | 0.7079 | 0.7097 | 0.6849 |
| Label-only Conditional Generative Data Augmentation | cDiff | 0.8034 | 0.8130 | 0.7921 | 0.8100 | 0.7833 | 0.7796 | 0.7660 | 0.7451 | 0.7374 | 0.7198 | 0.7225 | 0.7107 |
| | cVAE | 0.8016 | 0.8124 | 0.7928 | 0.8150 | 0.7655 | 0.7538 | 0.7720 | 0.7615 | 0.6982 | 0.6772 | 0.6979 | 0.6810 |
| | cGAN | 0.7531 | 0.7431 | 0.7221 | 0.7354 | 0.7322 | 0.7110 | 0.7082 | 0.6885 | 0.6721 | 0.6630 | 0.6514 | 0.6330 |

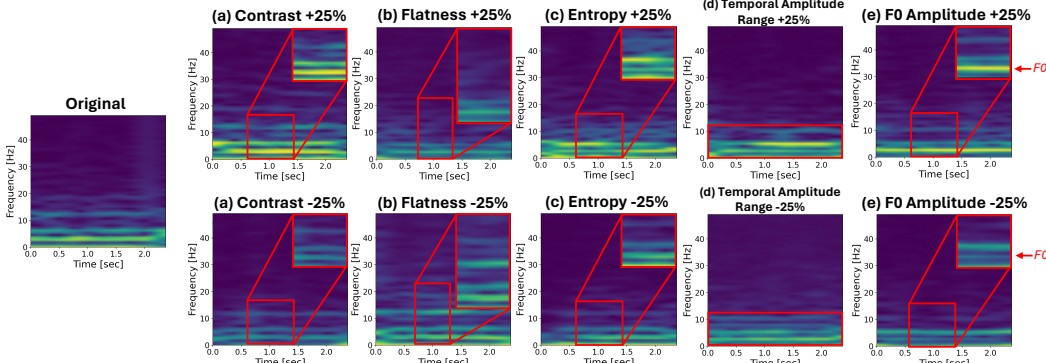

Figure 4: **Visualization of Metric Control Effects on Accelerometer Spectrogram Generation from HAR Dataset**. The inherent label of this signal is "running". We altered a single metric condition while retaining the remaining metrics. **(a)** Increasing contrast enhances the distinction between peak and valley values, while decreasing contrast reduces the differences, making high and low energy areas less pronounced. **(b)** Higher flatness results in a uniform and consistent energy distribution across the spectrum; lower flatness leads to pronounced interleaving between peaks and valleys, exhibiting stronger harmonic patterns. **(c)** Higher entropy creates a more random or chaotic energy distribution, whereas lower entropy results in a simpler and more predictable spectrogram appearance. **(d)** Increasing the temporal amplitude range introduces greater dynamics over time, whereas reducing it leads to a more uniform amplitude distribution throughout the timeline. **(e)** Adjusting the F0 amplitude emphasizes the intensity of the fundamental frequency without affecting the relative amplitudes of other harmonics. Additional visualizations are available in Appendix G.

realistic compared to those produced by the diffusion model, a finding supported by numerous previous studies [28, 2]. We hypothesize that the inferior generative quality of the VAE limits the effectiveness of data augmentation. This suggests that the success of our data augmentation method depends significantly on the underlying generative model's ability to synthesize authentic data.

To validate the fine-grained control over the generation process, we visualize examples of the generation results in Figure 4. Results demonstrate the precise fine-grained control of our approach over the IoT sensing signal generation.

### 4.3 Data Augmentation with Domain Knowledge Instruction

We highlight the effectiveness of integrating domain knowledge to enhance data augmentation through three case studies. In each case, we create a domain shifting scenario and aim to increase the downstream model's robustness to variations outside the training set. We utilize the diffusion model as the generative model backbone and DeepSense as the downstream model.

**Case study 1: Various Running Styles in HAR dataset**. We use the RealWorld-HAR dataset for our analysis. For the training set, we select 3 human subjects who demonstrated cautious style during the data collection of the "running" activity. For the validation set, we choose other 3 human subjects who exhibited energetic running patterns. The distinct running patterns of the subjects were assessed by reviewing video footage recorded by an observer holding a camera and following the runners. [2] Based on IoT sensing domain knowledge, we hypothesize that the more forceful arm swings of energetic runners produce more intense and varied acceleration signals. The dominant arm swing frequency and its harmonics are also more pronounced. In spectrograms, these characteristics should manifest as higher contrast, lower flatness, and stronger F0 amplitude compared to the cautious runners. As shown in Figure 5a, our visual inspections of accelerometer spectrograms from each participant confirmed these assumptions.

We then simulate a scenario where we only have the data from the 3 cautious runners, and need to generalize the activity recognition to the running activity of the energetic runners. In our training set, we have the data of all activities from the cautious runners, while the validation data comprises running activities from the energetic runners. Leveraging our domain insights, we directionally augment the data by intensifying the contrast and fundamental frequency amplitude and reducing the flatness in the spectrograms of the cautious runners. We set the varying range of contrast, flatness, and fundamental frequency amplitude at [10%, 40%], [-40%, -10%], and [10%, 40%] respectively, while keeping the other metrics unchanged. Having these ranges, we employ the perturbation augmentation method to create the augmented data. In Figure 5b, results show that our domain knowledge instruction (noted as *knowledge*) substantially improves the results, closely approaching the upper performance bound set by *Reconst*. This demonstrates that the generated data can more precisely reflect the true characteristics of the diverse running styles.

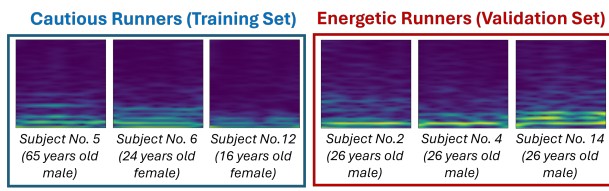
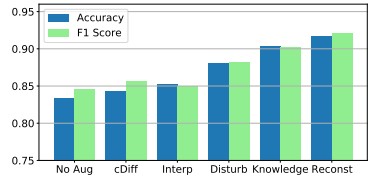

(a) Visual Comparison of Spectrograms between Cautious and Energetic Runners.

(b) Performance Comparison.

Figure 5: **Case Study 1.** Leveraging our insights into different running styles, our method enables more precise data augmentation and enhances the downstream model's performance on "running" activity recognition.

**Case Study 2: Various Road Types in Vehicle Detection Dataset**. In the vehicle detection dataset, the same vehicle driving on different road types exhibit various signal patterns. Our domain expertise suggests that the uneven and shifting surface of a gravel road generates sound and vibration profile with higher entropy compared to smoother concrete surface. As presented in Figure 6a, for the same vehicle type, spectrograms exhibit more complex and disordered patterns when on gravel than on concrete.

In this case study, we simulate a scenario where the training set only includes data collected on the concrete road while the validation set is the data collected on the gravel ground. By applying our knowledge about the effect of gravel surfaces, we generate synthetic data by increasing entropy and flatness and decreasing contrast. We set the varying range of these 3 metrics as [10%, 30%], [10%, 30%], and [-30%, 0] respectively, while keeping the other metrics unchanged. Then we use perturbation augmentation to synthesize the data. In Figure 6b, the results show that our data

---

[2]Public information and video footage about the human subjects can be found at: https://www.uni-mannheim.de/dws/research/projects/activity-recognition/dataset/dataset-realworld/

augmentation informed by precise domain knowledge, significantly enhances the generalizability of the downstream model from concrete road to gravel road.

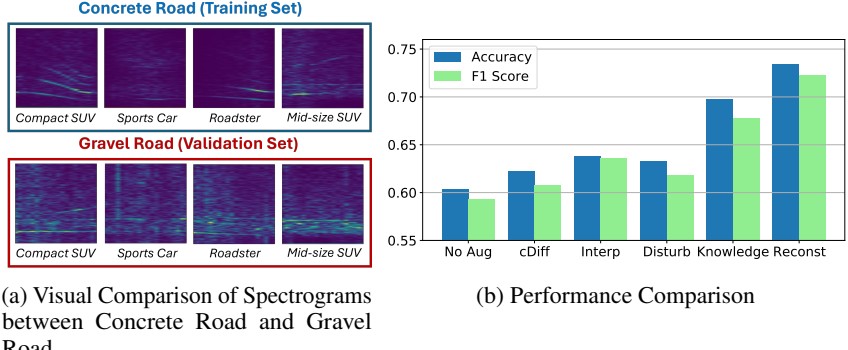

(a) Visual Comparison of Spectrograms between Concrete Road and Gravel Road.

(b) Performance Comparison

Figure 6: **Case Study 2.** Enhanced by our method with domain knowledge instruction, the downstream model more effectively manages domain shift in road types.

**Case Study 3: Various Seizure Patterns in Harmful Brain Activity Dataset**. We observe that the seizure signals from different patients can be divided into two categories regarding the fundamental frequency patterns. One is with clear fundamental frequency, which might be associated with the repetitive firing patterns of neurons in a specific brain region. We call it *Clear Seizure*. The other category has chaotic, noisy signals and blury fundamental frequency, which might be related to different brain regions contribute varying frequencies and patterns simultaneously. We call it *Chaotic Seizure*. In this case study, we manually select 10 patients of clear seizure as the traing set, and select 10 patients of chaotic seizure as the validation set. To augment the training data, we set the varying range of contrast, entropy, and fundamental frequency amplitude at [10%, 25%], [10%, 40%], and [-25%, -10%] respectively, while keeping the other metrics unchanged. We show example spectrograms in Figure 7a. The experiment results are shown in Figure 7b. Among all the baselines, our domain knowledge instruction achieves the best performance, again proving the effectiveness and generalizability of our approach.

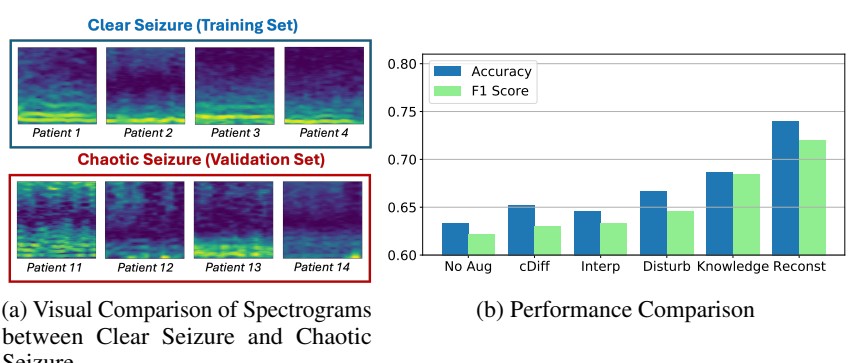

(a) Visual Comparison of Spectrograms between Clear Seizure and Chaotic Seizure.

(b) Performance Comparison

Figure 7: **Case Study 3.** With domain knowledge instruction, our model achieves the best on bridging the gap between seizure patterns between two groups of patients.

## 4.4 Number of Augmentation Data

We investigate the influence of the number of augmentation data on the downstream model performance. Here we use diffusion model as the generative model backbone, and DeepSense as the downstream model. As depicted in Figure 8, even though the ideal augmentation ratio varies in different datasets, a steep climb in performance occurs from 50% to 500%. Among the 3 datasets, human activity recognition has the smallest data volume, which is below 5000 training samples. We envision that the generative capability is fully expressed at a lower augmentation ratio. Harmful brain

activity recognition dataset has the largest number of training samples ($\approx$ 80,000), where a tendency of performance growing remains even at a higher augmentation ratio.

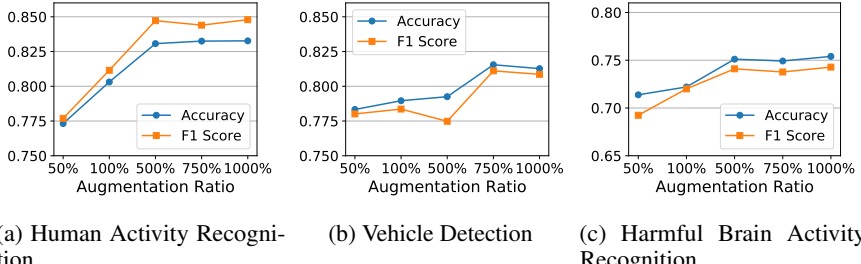

(a) Human Activity Recognition

(b) Vehicle Detection

(c) Harmful Brain Activity Recognition

Figure 8: **Impact of Augmentation Ratio on Downstream Model Performance.** Augmentation ratio = Number of generated samples / Number of real samples in the training set

## 4.5 Ablation Studies

Table 2: **Ablation Studies.** Results with 95% confidence interval.

| Metrics | Human Activity Recognition | | Vehicle Detection | | Harmful Brain Activity Recognition | |
|---|---|---|---|---|---|---|
| | Acc | F1 | Acc | F1 | Acc | F1 |
| All metrics | 0.8307±0.0171 | 0.8333±0.0145 | 0.7925±0.0053 | 0.7748±0.0095 | 0.7511±0.0116 | 0.7410±0.0143 |
| No time domain metric | 0.8290±0.0202 | 0.8305±0.0212 | 0.7745±0.0122 | 0.7691±0.0104 | 0.7233±0.0295 | 0.7058±0.0268 |
| No frequency domain metric | 0.8177±0.0155 | 0.8208±0.0166 | 0.7910±0.0068 | 0.7554±0.0066 | 0.7225±0.0151 | 0.7189±0.0144 |
| No time-frequency domain metric | 0.8112±0.0166 | 0.8146±0.0120 | 0.7880±0.0173 | 0.7753±0.0199 | 0.7133±0.0103 | 0.7021±0.0097 |

To investigate the impact of different metrics on our data augmentation approach, we ablate each category of metrics and evaluate their effects using the disturb augmentation method with diffusion model as the generative model backbone. Performance was evaluated using DeepSense as the downstream model. As depicted in Table 2, excluding any metrics consistently results in diminished performance, validating the effectiveness of our selected metrics. The results also indicate varying importance of metrics across different datasets.

## 5    Conclusion

In this paper, we present a method of fine-grained control on generative augmentation in IoT sensing applications. Our novel data augmentation technique merges the advantages of leveraging domain knowledge specific with the production of highly authentic synthetic samples. Compared to traditional transformation-based and generative data augmentation methods, our approach demonstrates superior performance, particularly when domain-specific knowledge is available, which is a common scenario in many IoT sensing tasks. Furthermore, the versatility of our method enables its application across different generative models, allowing it to benefit from the inherent generative capabilities of these models.

## Acknowledgments

Research reported in this paper was sponsored in part by DEVCOM ARL under Cooperative Agreement W911NF-172-0196, NSF CNS 20-38817, and the Boeing Company. It was also supported in part by ACE, one of the seven centers in JUMP 2.0, a Semiconductor Research Corporation (SRC) program sponsored by DARPA. The views and conclusions contained in this document are those of the authors and should not be interpreted as representing the official policies, either expressed or implied, of the Army Research Laboratory, DARPA, or the U.S. Government. The U.S. Government is authorized to reproduce and distribute reprints for Government purposes notwithstanding any copyright notation herein.

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

## Appendix

## A   Limitations

**Metric Selection:**  One major limitation of our current approach is that the metrics are selected empirically rather than through a quantitative method. This means our choices are based on observation and experience on the limited IoT sensing datasets, which might introduce bias or overlook potential correlations within certain metrics and the data. Also, when new sensing applications are involved or sensing modalities are introduced, the functionality of these metrics need to be empirically validated again, which is inefficient. We envision the development of an improved method for assessing metric importance to mitigate this issue. For instance, we could identify a broader set of potential metrics and analyze their correlations with the discriminative features identified by the downstream model. The strength of these correlations could then be used to rank the importance of each metric among all candidates.

**Introduction of Domain Knowledge:**  Even after we obtained the fine-grained control over the generation process, deciding the proper augmentation strategy still requires deliberate considerations. The introduction of domain knowledge is not straightforward and involves a nuanced understanding of the underlying mechanisms of the target domain. Currently, this process relies heavily on the expertise of practitioners who understand both the technical aspects of generative models and the specific characteristics of the domain. This is an issue that pervasively exist in all the domain knowledge-guided neural network design[8, 27]. We believe this is highly challenging and we look forward to make advancements in future research.

## B   Metrics Design

We formally define the statistical metrics we proposed in constructing the metric space.

Let $S \in \mathbb{R}^{f \times t}$ represent the STFT spectrogram, where $f$ and $t$ are the dimensions of frequency and time, respectively.

### B.1   Temporal Amplitude Range

The temporal amplitude range measures the variability in signal amplitude across time in a spectrogram after averaging over the frequency dimension. It is defined mathematically as:

$$\textit{Temporal Amplitude Range} = \max_{t} \left( \overline{A}(t) \right) - \min_{t} \left( \overline{A}(t) \right) \tag{4}$$

where $\overline{A}(t) = \frac{1}{f} \sum_{i=1}^{f} S(i, t)$ represents the average amplitude at time $t$ over all frequency bins $f$ in the spectrogram $S$. This metric captures the total temporal variation in signal strength and provides insights into the operational limits of the IoT sensing signal.

### B.2   Fundamental Frequency (F0) Amplitude

We define the fundamental frequency amplitude as the mean amplitude at the fundamental frequency across all time bins. To calculate that, we first estimate the fundamental frequency $f_0$ for each time frame in the spectrogram using the Harmonic Product Spectrum (HPS) method.

This process involves the following 3 steps:

**Step 1: Compute the HPS for Each Time Frame**

For each time frame $t$ in the spectrogram $S(f, t)$, the HPS is computed by:

1. Generating downsampled versions of the spectrum for the given time frame, where the downsampling occurs on the frequency dimension only:
$$X_d(f, t) = S(f \downarrow d, t) \quad \text{for } d = 1, 2, \ldots, D \tag{5}$$
   where $f \downarrow d$ denotes downsampling the frequency index $f$ by a factor of $d$, and $D$ is the maximum downsampling factor. This approach reduces the frequency resolution while maintaining the temporal resolution intact.

In practice, we set $D$ for various datasets to accommodate different sampling rates and harmonic patterns of the modalities. Specifically, for the HAR dataset, $D = 2$ is used for both accelerometer and gyroscope signals. In the vehicle detection dataset, $D = 3$ is set for acoustic signals and $D = 2$ for seismic signals. For harmful brain activity recognition, $D = 2$ is applied to EEG signals.

2. Multiplying these downsampled spectra together to emphasize the fundamental frequency:

$$H(f, t) = \prod_{d=1}^{D} X_d(f, t) \tag{6}$$

**Step 2: Identify $f_0$ for Each Time Frame**

The fundamental frequency $f_0(t)$ at each time frame $t$ is identified as the frequency corresponding to the maximum value in $H(f, t)$:

$$f_0(t) = \arg\max_f H(f, t) \tag{7}$$

**Step 3: Calculate Average F0 Amplitude**

Once the fundamental frequencies are determined for each time frame, the average amplitude at these frequencies is computed by averaging the amplitudes at $f_0(t)$ over all time frames:

$$\textit{F0 Amplitude} = \frac{1}{T} \sum_{t=1}^{T} S(f_0(t), t) \tag{8}$$

where $T$ is the total number of time frames.

## B.3 Contrast

Contrast calculates the distance between the amplitude peaks and valleys in the spectrogram. We define the sets of peaks and valleys:

- Let $P$ be the set of values in $S$ that are in the top 5% of all values in $S$.

$$P = \{S(i, j) \mid S(i, j) > \text{quantile}(S, 0.95)\} \tag{9}$$

- Let $V$ be the set of values in $S$ that are in the bottom 5% of all values in $S$.

$$V = \{S(i, j) \mid S(i, j) < \text{quantile}(S, 0.05)\} \tag{10}$$

Contrast is defined as:

$$Contrast = \frac{1}{|P|} \sum_{p \in P} p - \frac{1}{|V|} \sum_{v \in V} v \tag{11}$$

Where $|P|$ and $|V|$ represent the number of elements in the sets $P$ and $V$, respectively.

## B.4 Flatness

We first define the power spectral density, denoted as $P(f, t)$, of the spectrogram $S(f, t)$:

$$P(f, t) = |S(f, t)|^2 \tag{12}$$

Flatness of a spectrogram is defined as:

$$Flatness = \frac{\exp\left(\frac{1}{TF} \sum_{t=1}^{T} \sum_{f=1}^{F} \log P(f, t)\right)}{\frac{1}{TF} \sum_{t=1}^{T} \sum_{f=1}^{F} P(f, t)} \tag{13}$$

where $T$ is the total number of time bins, $F$ is the total number of frequency bins, and $TF$ represents the total number of elements in the spectrogram.

### B.5 Entropy

To calculate entropy, we first normalize the spectrogram to create a probability distribution:

$$p(f,t) = \frac{S(f,t)}{\sum_{f=1}^{F}\sum_{t=1}^{T} S(f,t)} \tag{14}$$

Here, $p(f,t)$ represents the probability of observing the power value at time $t$ and frequency $f$.

The entropy of the spectrogram is then defined as:

$$Entropy = -\sum_{f=1}^{F}\sum_{t=1}^{T} p(f,t) \log p(f,t) \tag{15}$$

where the sum is taken over all time and frequency bins, and the logarithm is base 2.

## C  Datasets

We describe the details of each dataset in the following.

### C.1  Human-Activity Recognition (HAR)

We utilize the Real-World HAR dataset [47], which consists of synchronized readings from multiple wearable sensors on 15 human subjects. Specifically, we select data from the accelerometer and gyroscope mounted on the upper arm, both of which record at a 100 Hz sampling rate. Each subject performed five dynamic activities for approximately 10 minutes each, except for jumping, which lasted about 1.7 minutes. These activities include climbing down, climbing up, jumping, running, and walking. We divide the dataset randomly by subjects, assigning 10 subjects to the training set and the remaining 5 to the validation set. We segmented the entire recording into 2.5-second segments. This partition results in 4,959 training samples and 2,309 validation samples. Due to the individual physical variations among subjects during the same activity, a domain shift occurs when the validation dataset comprises human subjects not included in the training set.

### C.2  Vehicle Detection

We created our own dataset by deploying 5 IoT sensing nodes, each equipped with a seismic sensor (100 Hz sampling rate) and an acoustic sensor (800 Hz sampling rate), placed on the ground to monitor the sounds and vibrations from passing vehicles. The goal is to develop a model that can classify vehicle types based on this sensor data, with each segment being 5 seconds long. The data was collected across 3 different locations: a city parking lot with a concrete surface, a state park parking lot with a gravel surface, and a country road with a dirt surface. This dataset includes the footage of 5 different vehicle types: a compact-size SUV, a mid-size SUV, a sports car, and a convertible roadster. Each data collection session lasted for 40 minutes, during which a single vehicle type traversed the testing field at speeds between 5 and 30 miles per hour along a random path. The initial 30 minutes of data are used for training, while the final 10 minutes serve as validation. Data segments where the vehicle was more than 65 feet from any sensor were excluded based on GPS distance calculations. This process results in 8,015 training samples and 2,763 validation samples.

### C.3  Harmful Brain Activity Recognition

We adopt Harmful Brain Activity Classification [18, 19] dataset which comprises EEG signals obtained from hospitalized patients who underwent EEG monitoring as a part of their clinical care. The sampling rate of the EEG sensors is 100 Hz. Each original recording, approximately 10 minutes long, is segmented into 40-second intervals. The objective is to classify 5 types of harmful brain activities using these data segments, including seizure (SZ), generalized periodic discharges (GPD), lateralized periodic discharges (LPD), lateralized rhythmic delta activity (LRDA), and generalized rhythmic delta activity (GRDA). The ground truth labeling was done by human experts manually[20]. We allocate 80% of the original EEG recordings to the training set and 20% to the testing set. This partition results in 78,548 training samples and 19,532 validation samples.

# D  Generative Models

## D.1  Diffusion Model

Our diffusion model utilizes the architecture proposed in [42], which incorporates the U-Net as the backbone [41], a design prevalent across many diffusion models. U-Net comprises multiple convolutional layers, pooling, and up-sampling operations, all interconnected through skip connections that maintain spatial hierarchies for detailed segmentation. When a 2D spectrogram gets into the U-Net, it is first progressively downsampled through convolutional and pooling layers to extract features, and then incrementally upsampled using transposed convolutions. Skip connections integrating details from corresponding downscaled layers to the upscaled layers for precise localization. In our implementation, we have 3 layers for downsampling and 3 layers for upsampling. Each downsampling/upsampling rescales both the time and frequency dimension of the spectrogram by 2. We set the intermediate channel dimension as 128 for the HAR and vehicle detection datasets, while 64 for the harmful brain activity recognition dataset. Both the HAR and vehicle detection datasets have 45.5 million parameters, while the harmful brain activity recognition dataset has 11.32 million.

Our conditioning embedder begins by taking each metric value $m_i$ with a channel dimension $d_i$, reshaping it to $d_i \times 1 \times 1$. This reshaped tensor is then replicated $T$ times along the third dimension to align with the spectrogram's time dimension. The tensor subsequently passes through a individual linear layer which increases its channel dimension to $F$, altering its shape to $d_i \times F \times T$. This process is consistently applied to each metric value and the one-hot embedding of the data label. Following this, all transformed tensors are stacked together on the channel dimension, resulting in a tensor of dimensions $\sum_i^M d_i \times F \times T$, where $M$ represents the total number of metrics and the label. We then compute the average over the first dimension of this stacked tensor, yielding a final dimension of $1 \times F \times T$. This resulting tensor, denoted as $\mathbf{c}$, serves as the condition tensor and is concatenated along the channel dimension with either the input spectrogram $\mathbf{x}$ or noise $\epsilon$, which is in shape of $D \times F \times T$, where $D$ is the feature channel of the sensor signals. The concatenated tensor is in shape of $(D+1) \times F \times T$, and then fed into the U-Net described above.

During the training of the diffusion model, we compute the metric loss using the generated spectrograms. To accelerate the generation process, we employ a diffusion step of 10, which offers a balance between moderate quality and rapid generation. For the data generation phase, however, we utilize a diffusion step of 100 to produce augmented data of higher quality.

We train the diffusion model using an Adam optimizer with a learning rate of 0.0001, paired with a cosine annealing learning rate scheduler. The model is trained for 1,000 epochs on the HAR and vehicle detection datasets, and for 200 epochs on the harmful brain activity recognition dataset. The batch size is set at 200 for the HAR and vehicle detection datasets, and at 64 for the harmful brain activity recognition dataset.

We handle the multi-modal inputs from HAR (accelerometer + gyroscope) and vehicle detection (acoustic + seismic) by employing two separate U-Nets, each dedicated to processing one modality. Although we anticipate strong correlations between the modalities, no modality fusion occurs within between the two U-Net architectures. We consider integrating modality fusion as a potential direction for future research to enhance the diffusion model's performance with multi-modal IoT sensing data.

Our model was trained on a desktop with Intel(R) Core(TM) i9-9960X CPU @ 3.10GHz and 4 Nvidia GeForce RTX 2080 Ti. The training of the diffusion model on a single GPU lasts for around 2 days for HAR and vehicle detection datasets, and around 5 days for harmful brain activity recognition dataset.

## D.2  VAE

VAEs have been extensively applied to various IoT tasks in prior research. [45, 26, 59, 62]. To capture the hierarchical features of the spectrogram, our VAE model is developed based on the NVAE [51]. The effectiveness of such hierarchical structure has been tested and verified in multiple prior studies [7, 15, 17]. The original NVAE model constitutes of a 3-layer encoder and a 3-layer decoder, where each layer is a residual cell, in order to model the long-range correlations in data[51]. Unlike the naive VAE model [24], NVAE designs its reparameterization process as a 3-step dependent sampling where each step is completed right before each layer of the decoder. Our conditional NVAE model remains

most of the architecture and settings of the original NVAE with only a few changes. We replace the prior trainable parameters in the first layer of NVAE's decoder with our condition embedder, following the same settings used in our diffusion model. Similarly, we also add a condition embedder at the encoder to concatenate our conditioning embedding to the input data.

During the training process, we feed our conditions and ground truth data to the encoder to generate the latent variables of each layer. Then we feed the conditions to the beginning of the decoder and the latent variables to each layer's sampling process respectively, to get the generated data. We calculate loss as the mean squared error (MSE) between the ground truth and generated data, added by a weighted KL divergence loss. The weighting for each dataset is as follows: 1 for the HAR dataset, 10 for the vehicle detection dataset, and 8 for the harmful brain activity recognition dataset.

We train the NVAE model using an Adam optimizer with a learning rate of 0.0005, paired with a cosine annealing learning rate scheduler. Our model is trained for 200 epochs on the HAR dataset, 300 epochs on the vehicle detection dataset, and 600 epochs on the harmful brain activity recognition dataset. The bacth size of all datasets is set as 64.

The total number of parameters for each dataset is 19 million. We trained our VAE model on the same machine as our diffusion model. The training time of each dataset is 1 hour for the HAR dataset, 2 hours for the vehicle detection dataset, and 16 hours for the harmful brain activity recognition dataset.

## E   Baselines

**Data Space Interpolation**: We follow the method called *mixup* proposed in [63]. Mixup conducts interpolation within pairs of examples and their labels. The intuition is that mixup regularizes the neural network to favor simple linear behavior in-between training examples, which reduces undesirable oscillations when predicting outside the training examples.

Given a pair of randomly selected data samples $(x_i, y_i)$ and $(x_j, y_j)$, where $x_i$ and $x_j$ are data vectors, while $y_i$ and $y_j$ are one-hot label encodings, the interpolated data sample $(x', y')$ is constructed by:

$$\begin{aligned} x' &= \lambda x_i + (1 - \lambda)x_j, \\ y' &= \lambda y_i + (1 - \lambda)y_j \end{aligned} \tag{16}$$

In our implementation, we follow the original authors' setup by randomly sampling $\lambda$ from a beta distribution:

$$\lambda \sim Beta(0.2, 0.2) \tag{17}$$

**Feature Space Interpolation:**   As suggested by [10], we first train an autoencoder using an unsupervised approach, followed by a linear interpolation of the intermediate feature maps from two randomly chosen samples of the same activity to generate synthetic feature maps. The architecture of the autoencoder largely follows our VAE design described in Appendix D.2, but only keep the reconstruction loss during the training. These synthetic alongside the realistic feature maps are subsequently utilized to train the downstream models.

**Time and Frequency Domain Transformation:**   To augment a data sample, we first randomly apply 2 out of 7 time domain transformations to the data in time series form. Subsequently, we employ short-time Fourier transform to convert the time-series data into 2D spectrograms, and apply 1 out of 2 frequency domain transformations. We adopt the time domain transformations described in [50], including jittering, scaling, negation, permutation, time warping, magnitude warping, rotation, and cropping. For frequency domain transformations, we incorporate frequency masking[38] and frequency perturbation[11].

**Conditional Diffusion Model:**   This diffusion model has the same design depicted in Appendix D.1, with the distinction that the condition vector comprises only the embedding of the inherent label. During the generation phase, synthetic samples for each class are created by the trained diffusion model using the respective class label as conditions. This approach facilitates a direct comparison to assess the benefits of the fine-grained control and metric space augmentation we propose.

**Conditional VAE:**   This method follows the design described in Appendix D.2, but differs in that the condition vector includes only the embedding of the inherent label. This setup is designed to evaluate the impact of label embedding on the generation process within the VAE framework.

**Conditional GAN:** We utilize the deep convolutional generative adversarial network (DCGAN) architecture[40] for this model. To enhance the stability and balance between the generator and discriminator, we implement optimization strategies on the loss terms as suggested by [1]. Similar to the other models, this GAN also uses only the inherent label as a condition for generation

# F    Downstream Models

**DeepSense:** [61] This is a deep neural network tailored for handling time series data from the IoT sensing applications. This network operates in the time-frequency domain, transforming the data using the short-time Fourier transform. To process data from multi-modalities, the model includes two separate convolutional layers for each sensor, with the resulting intermediate feature maps being merged subsequently. After that, it employs 2 2D-convolutional layers to extract spatial features from the data. Following this, 1 Gated Recurrent Unit (GRU) layer is used to learn temporal patterns. Next, 2 linear layers are used for refining the feature dimensions and generate the final classification logits.

**Transformer (Encoder-Only):** [52] In the Transformer network, each input spectrogram from different modalities is fed into a standard Transformer encoder layer, incorporating a self-attention layer along with two linear layers. Here, the attention is applied to along the time dimension, while the frequency and feature dimensions are multiplied, serves as the embedding dimension. Multimodal features are subsequently integrated through concatenation followed by processing via a linear layer. The classification mechanism at the end consists of a linear layer followed by a softmax function.

# G    Additional Visualization

## G.1    Diffusion Model vs. VAE

We compare the generation quality of our method when using the diffusion model versus the VAE as the generative model. For this analysis, we use accelerometer signals from the human activity recognition dataset. Synthetic samples are generated under the same conditions as the original data samples. We select two groups of the original spectrograms based on their complexity in patterns: the first group has simpler and more pronounced patterns, while the second group has more complex and chaotic patterns.

As illustrated in Figure 9, the VAE-generated results are comparable to those from the diffusion model in the first group of the selected samples. Both the diffusion model and VAE successfully capture the fundamental frequency and its harmonic patterns. This group of selected samples exhibits temporal consistency with pronounced harmonic patterns that are straightforward to replicate. However, despite the relatively simple structure of these realistic samples, the spectrograms produced by the VAE are less detailed than those from the diffusion model, particularly in the higher frequency regions.

For the second group, which features more complex original spectrograms as shown in Figure 10, the generation quality of the VAE significantly degrades. The findings indicate that the VAE struggles to replicate the intricate textures present in the original spectrograms, whereas the diffusion models continue to effectively capture and reflect these characteristics.

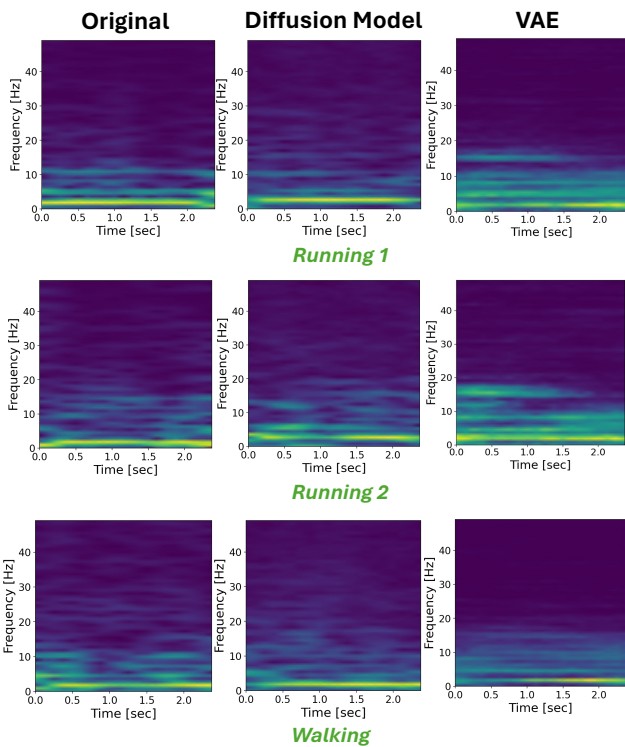

Figure 9: **Simpler Cases of Generation.** The original spectrograms exhibit strong and consistent patterns that are easier to capture. Both models capture the fundamental frequency and harmonics. But VAE exhibits more blurry patterns.

The observation that VAE generates data with lower quality has been recognized in many prior research[28, 2]. Consistent with these findings, our experiments also demonstrate that the VAE, in its current implementation, exhibits lesser generative capabilities compared to our diffusion model.

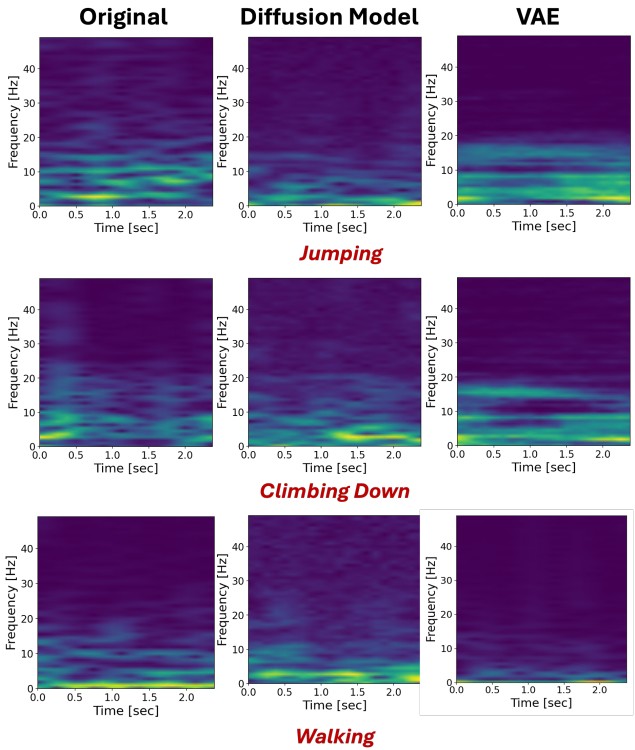

Figure 10: **More Complex Cases of Generation.** The original spectrograms are more chaotic and textured. VAE fails in capturing those details and variations.

## G.2 Visualization of Vehicle Detection Dataset

In Figure 11, we randomly select an acoustic sample from the vehicle detection dataset and visualize the generated spectrograms when manipulating the metrics as conditions. Again, we change a single metric condition while retaining the remaining metrics to show to control effects.

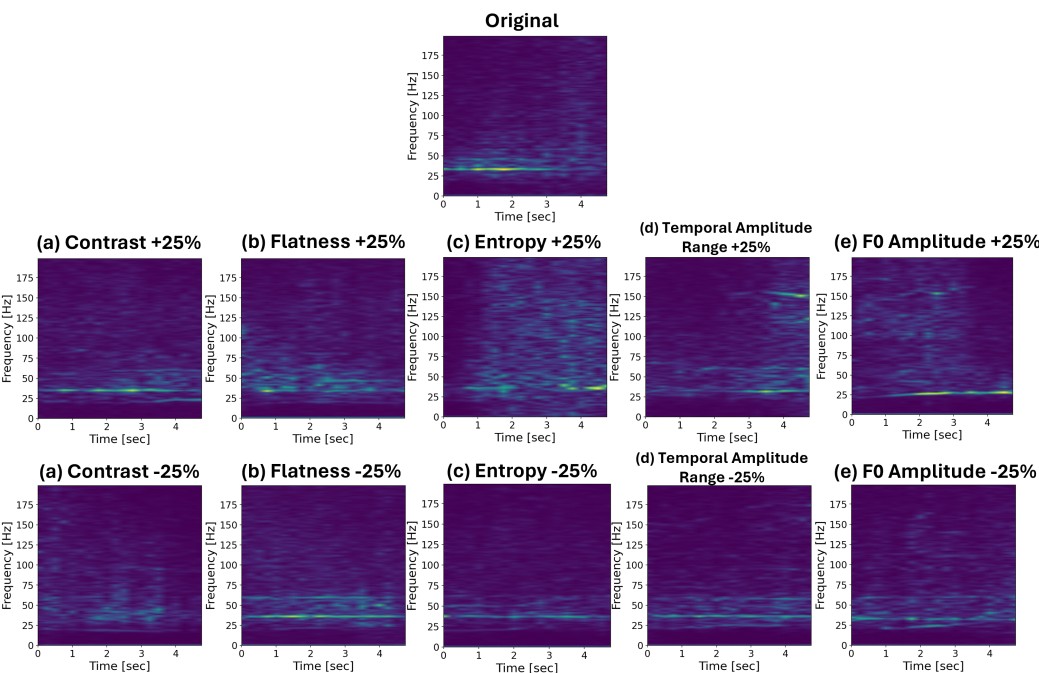

Figure 11: Visualization of Metric Control Effects on Acoustic Spectrogram Generation from Vehicle Detection Dataset.

## G.3  Visualization of Harmful Brain Activity Recognition

In Figure 12, we randomly select an EEG sample from the harmful brain activity recognition dataset and visualize the generated spectrograms. Like above, when manipulating a certain metric condition, we keep the remaining metrics unchanged.

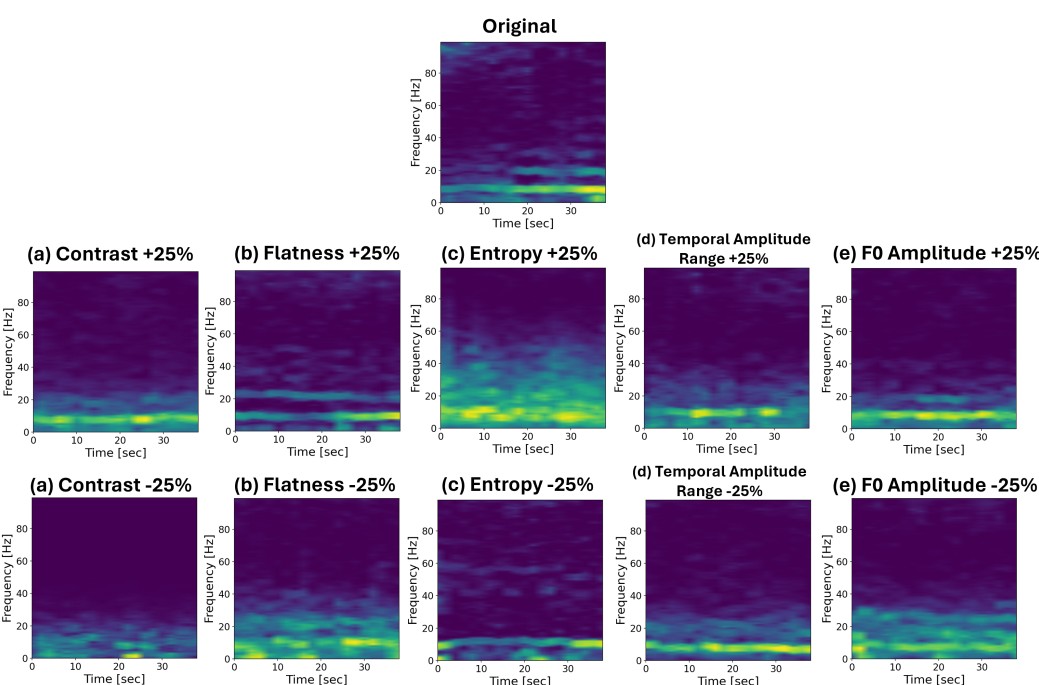

Figure 12: Visualization of Metric Control Effects on EEG Spectrogram Generation from Harmful Brain Activity Recognition Dataset.

