# OpenReview forum: "Fine-grained Control of Generative Data Augmentation in IoT Sensing"
_NeurIPS.cc/2024/Conference — NeurIPS 2024 poster_

### Official Review · Reviewer_eQAt · 2024-06-21

**Soundness:** 3
**Presentation:** 3
**Contribution:** 3
**Rating:** 6
**Confidence:** 5

**Summary:**

This work introduces fine-grained control to generative models for IoT data augmentation by defining a metric space with statistical metrics.

These metrics are essential for capturing the key features of the spectrograms and serve as strong constraints for the generative model, allowing for customization of spectrogram characteristics in the time-frequency domain tailored to specific application requirements.

The evaluation performs better than traditional transformation-based data augmentation techniques and previous generative data augmentation models.

**Strengths:**

1. A novel constraint design for generative models by defining a metric space with statistical metrics.

2. An interesting metric space augmentation method is proposed.

**Weaknesses:**

1. How to balance the original generative model loss and the designed metric loss is not well discussed. As mentioned by the authors, the two loss terms are ratioed by 10:1. It seems the metric loss contributes marginally during the generation process. Considering the generation process is a probabilistic process, without deep analysis, it is hard to convince readers that high-quality outputs are random or consistent.

2. The metrics selection is not well-founded. Though the authors have discussed its limitation in the metric selection, the metrics are data-type and task-oriented. The inappropriate selection of the metrics could introduce extra noise. Which or how these metrics contribute to the better generative process should be further analyzed. Overall, the contribution is limited as in-depth analysis is missing, and complex hyperparameter (e.g., metric selection, weight selection) tunning is involved.

3. The evaluation setting is trivial, which can hardly demonstrate the efficacy of high-quality data generation. For example, the HAR dataset has only 5 simple activities, which is not a data-hungry task. The authors are encouraged to check more complex real-world datasets, considering the data and label heterogeneity.

**Questions:**

1. What kind of outputs will be if we choose different weights between metric loss and generative loss? It seems this is not an easy job to adjust every weight corresponding to each metric according to the loss during the model training process.

2. Why do not authors choose other metrics? Are the selected metrics general to all IoT tasks? What if some data types are not sensitive to the proposed metrics?

3. How large the dataset is required to train the diffusion model for the generation task?

4. Is it possible the interpolation, and disturbance in the metric space change the metric to respond to a different label, which conflicts with the original label? If so, can authors provide such low-quality generation examples?

5. Will there be a domain shift between generated samples and raw samples?

6. Could authors provide the magnitude of the spectrogram?

**Limitations:**

The authors have discussed the limitations in metric selection and the introduction of domain knowledge.

The authors have kindly answered my questions and I appreciate their efforts for the research community.

---

> ### Author Rebuttal · Authors · 2024-08-07
>
> ### **Author Reply to Reviewer eQAt (Part 1/3)**
>
> Thank you for your approvals, critiques and insightful questions! We would like to address your concerns and provide extra evaluation results as follows.
>
> > W1: How to balance the original generative model loss and the designed metric loss is not well discussed. As mentioned by the authors, the two loss terms are ratioed by 10:1. It seems the metric loss contributes marginally during the generation process. Considering the generation process is a probabilistic process, without deep analysis, it is hard to convince readers that high-quality outputs are random or consistent.
>
> **Response**: We set the ratio of the two loss terms to 10:1 by default, intending to prioritize the optimization of the generative model itself using $L_G$ in the early stages of training, and then gradually shift focus to enforcing control over the generation using $L_{metric}$ in the later stages. In Figure 12 in the ***uploaded pdf file***, we plot the curves of the two loss terms when training on the HAR dataset with ratios of 10:1 and 5:1, respectively. We observe that with a 10:1 ratio, $L_G$ decreases rapidly and eventually matches the scale of $L_{metric}$ after a few hundred epochs. Thus, the 10:1 ratio does not make the contribution of $L_{metric}$ trivial. We also notice that decreasing this ratio delays the intersection of the curves of the two loss terms, causing $L_{metric}$ to converge at a slightly higher value. Therefore, we fix the ratio at 10:1 in our implementation.
>
> We further profile the influence of weight ratio $L_G:L_{metric}$ on the performance as below. Due to time limit, we only show results for Human Activity Recognition dataset using Diff-disturb as the data augmentation method.
>
> *DeepSense:*
> |          | 1:1 | 5:1 | 10:1 | 20:1 |
> |----------|-----|-----|------|------|
> | Accuracy |  0.8211 | 0.8307 |**0.8324**| 0.8249  |
> | F1-score |  0.8209 | 0.8333 |**0.8422**| 0.8360  |
>
> *Transformer:*
> |          | 1:1 | 5:1 | 10:1 | 20:1 |
> |----------|-----|-----|------|------|
> | Accuracy | 0.8199  | 0.8105  |  0.8205 | **0.8280**   |
> | F1-score | 0.8239  | 0.8195  |  **0.8352** | 0.8333   |
>
> As shown in the results above, when $L_G / L_{metric}$ is between 1~20, the performance generally stays stable. These findings indicate the robustness of the loss term design.
>
> > W2: The metrics selection is not well-founded. Though the authors have discussed its limitation in the metric selection, the metrics are data-type and task-oriented. The inappropriate selection of the metrics could introduce extra noise. Which or how these metrics contribute to the better generative process should be further analyzed. Overall, the contribution is limited as in-depth analysis is missing, and complex hyperparameter (e.g., metric selection, weight selection) tunning is involved.
>
> **Response**: Before the widespread adoption of advanced deep learning models, researchers often used statistical metrics as features for conducting IoT sensing tasks [1][2][3][4]. The metrics selected in our paper have been widely adopted in previous works and are generally applicable across different modalities, sampling rates, and physical phenomena, making them versatile for IoT sensing tasks. While we acknowledge that inappropriate metric selection can introduce noise, our choice was guided by their proven utility in existing IoT sensing literature and our own evaluation across three different IoT tasks.
>
> Nevertheless, we understand the need for a more in-depth analysis of how these metrics contribute to the generative process. As part of our future work, we plan to explore this further. This exploration will aim to refine the framework and potentially reduce the complexity, ensuring a more streamlined and efficient application in diverse IoT scenarios.
>
> [1] Detection of posture and motion by accelerometry: a validation study in ambulatory monitoring
>
> [2] Activity Recognition from Accelerometer Data
>
> [3] Vibration-based Terrain Classification Using Support Vector Machines
>
> [4] Epileptic activity recognition in EEG recording
>
>
> > W3: The evaluation setting is trivial, which can hardly demonstrate the efficacy of high-quality data generation. For example, the HAR dataset has only 5 simple activities, which is not a data-hungry task. The authors are encouraged to check more complex real-world datasets, considering the data and label heterogeneity.
>
> **Response**: In IoT sensing, unlike in vision or language-related tasks, data scarcity presents a significant challenge. Obtaining large labeled datasets is often difficult and costly. Even though the HAR dataset features only five activities, the variance among individuals' behaviors introduces substantial variability. This introduces the potential domain shift between the training and validation sets, making the task data-hungry and challenging. Our proposed method can effectively generalize from small datasets and enhance downstream tasks demonstrates its high data efficiency. This capability shows the practical value of our method, particularly in scenarios where labeled data is limited, showcasing its potential for real-world IoT sensing applications.
>
>
> > Q1: What kind of outputs will be if we choose different weights between metric loss and generative loss? It seems this is not an easy job to adjust every weight corresponding to each metric according to the loss during the model training process.
>
> **Response**: Thank you for the question. As discussed in our response to **W1**, different weights between the two loss terms $L_G$ and $L_{metric}$ can slightly influence the convergence speed of each loss. However, our profiling results indicate that the performance generally remains stable within the tested weight ratio range.

---

> ### Author Response · Authors · 2024-08-07
> **Author Reply to Reviewer eQAt (Part 2/3)**
>
> > Q2: Why do not authors choose other metrics? Are the selected metrics general to all IoT tasks? What if some data types are not sensitive to the proposed metrics?
>
> **Response**: As discussed in our response to W2, the metrics selected in our paper have been widely used in previous literature as critical features for depicting signals in various IoT sensing applications. However, we acknowledge that there is a broader range of metrics that we have not incorporated. We recognize this as a limitation and plan to explore this area in future work. Specifically, we aim to consider additional potential metrics and develop a mechanism for building a custom metric set tailored to specific IoT sensing tasks. For example, we could construct a decision tree for the IoT task based on all potential metrics and select them using the impurity-based importance score of each metric.
>
> > Q3: How large the dataset is required to train the diffusion model for the generation task?
>
> **Response**: We list the number of data samples in each dataset as below:
> |          | Human Activity Recognition | Vehicle Detection | Harmful Brain Activity Recognition
> |----------|-----|-----|------|
> | Train |  4,959 |  8,015 |78,548 |
> | Validation | 2,309 | 2,763  | 19,532  |
> | Total | 7,268 | 10,778 | 98,080 |
>
> Both the human activity recognition and vehicle detection datasets are relatively small, especially compared to large language or vision datasets. However, our evaluations demonstrate that these datasets are sufficient to support the training of our diffusion model for the purpose of data augmentation. In IoT sensing tasks, where labeled data can be expensive to obtain, data efficiency in training generative models is crucial. The effectiveness of our approach on these small IoT datasets highlights its great data efficiency.
>
> > Q4: Is it possible the interpolation, and disturbance in the metric space change the metric to respond to a different label, which conflicts with the original label? If so, can authors provide such low-quality generation examples?
>
> **Response**: Indeed, low-quality generation can occur if augmentation techniques are not properly applied. Figure 14 in the **uploaded pdf file** illustrates poor generation results. In these examples, we selected a metric for each original signal and manipulated its value by multiplying it by 0, 10, and 100, resulting in highly unnatural metric values. These modified metrics were then used as conditions for the generative model to synthesize the corresponding spectrograms. We observed that these out-of-distribution metric values led to unrealistic generation results. This demonstrates that augmentation methods, including interpolation and disturbance, must be used appropriately. The interpolation and extrapolation capabilities of the generative model are not limitless, and the designed metric conditions must remain within a reasonable range. However, we did not observe any cases where a sample, after modification in the metric space, explicitly exhibit patterns under a different label in our tested IoT tasks.

---

> ### Author Response · Authors · 2024-08-07
> **Author Reply to Reviewer eQAt (Part 3/3)**
>
> > Q5: Will there be a domain shift between generated samples and raw samples?
>
> **Response**: As with any generative model, there will inevitably be a domain shift between generated samples and real samples, as the model can only approximate the natural data distribution rather than perfectly replicate it. This inherent limitation means that the generated samples may not fully capture all nuances of the original data. However, the primary goal of using these augmented samples is to introduce variety and expand the training set, thereby enhancing the robustness and generalizability of the downstream model. As long as the augmented samples cover a substantial range of the data's possible variations, they can provide valuable additional training data. This can help the downstream model better understand the underlying data distribution and improve its performance, even in the presence of a domain shift.
>
>
>
> > Q6: Could authors provide the magnitude of the spectrogram?
>
> **Response**: We list the mean and standard deviation of the magnitude of spectrograms by modalities in each dataset as below:
>
> |                             | Human Activity Recognition |                     | Vehicle Detection         |                     | Harmful Brain Activity Recognition |
> |-----------------------------|----------------------------|---------------------|---------------------------|---------------------|-------------------------------------|
> |                             | Accelerometer              | Gyroscope           | Accelerometer             | Microphone          | EEG                                 |
> | Mean                        | 24.8311                    | 5.3611              | 0.9683                    | 0.5752              | 0.7792                              |
> | Standard Variance           | 91.3622                    | 13.4403             | 2.0264                    | 1.6248              | 0.7153                              |
>
>
> The three datasets have various spectrogram magnitudes, illustrating the diversity of data formats in IoT sensing tasks. The positive results achieved with these datasets demonstrate the generalizability of our approach.

---

> > ### Comment · Reviewer_eQAt · 2024-08-08
> >
> > Thank you so much for your detailed reply. Another question is how the synthetic data affects the model training process as there are distribution shift issues. Is there a way to filter out low-quality synthetic samples based on your proposed approach?

---

> > > ### Author Response · Authors · 2024-08-09
> > >
> > > Thanks for your further question! When the metric values are controlled within a reasonable range, we observe that the generated spectrograms from our data augmentation approach are mostly visually authentic and experimentally effective to the downstream models. In this case, we envision that the domain shift is reflected more on the limited generative capability: the generated data only mirrors a subset of real-world variations, thus the downstream model still cannot perform perfectly in the whole validation set. This is especially true for **interpolation** augmentation method, as it is a more conservative augmentation, aiming at producing intermediate data samples within the known data distribution.
> > >
> > > For augmentation methods with higher freedom, including **disturbance** and **domain knowledge instruction**, to strictly prevent generated data with metric values that are out of the designed boundary, we propose a simple filtering method.
> > > Specifically, we filter the generated dataset by the designed metric value boundary. For example, if our intention was to manipulate the contrast between -20%~20%, any generated data samples with contrast out of this range comparing to its original data sample will be excluded. Such outliers could exist as the control over metrics is not strictly guaranteed but through conditioning. By filtering out these outliers, the augmented data samples are further refined on their statiscal qualities.
> > >
> > > We conduct a simple example to demonstrate the effectiveness of the filtering method. We adopt the same setting described in Section 4.3 Case study 1, where we apply **domain knowledge instruction**-based data augmentation to the human activity recognition task. We compare the number of augmented samples before and after the filtering, as well as the corresponding downstream model performance using DeepSense.
> > >
> > > |          | Unfiltered | Filtered |
> > > |----------|-----|-----|
> > > |Augmented Sample Number | 24,795 | 22,922 (-7.55%) |
> > > | Accuracy | 90.31%  | 91.22% (+0.91%)  |
> > > | F1 | 0.9027 | 0.9175 (+0.0148) |
> > >
> > > Our findings are twofold. First, the metric conditions are generally well controlled as only 7.55% of the augmented samples requiring filtering. Second, the performance of the filtered augmented dataset slightly outperforms the unfiltered one, demonstrating the potential benefits of this further refinement.

---

> > > > ### Author Response · Authors · 2024-08-10
> > > >
> > > > We really appreciate your dedication in reviewing our paper, which has greatly assisted us in the paper review. Also great thanks for raising the rating!

---

### Official Review · Reviewer_eCvh · 2024-07-08

**Soundness:** 2
**Presentation:** 2
**Contribution:** 2
**Rating:** 6
**Confidence:** 3

**Summary:**

The paper addresses the challenge of overfitting in Internet of Things (IoT) sensing models caused by data distribution shifts between training datasets and real-world scenarios. To enhance model robustness, the authors introduce a novel data augmentation paradigm that adds fine-grained control to generative models. By defining a metric space with statistical metrics that capture the essential features of short-time Fourier transformed (STFT) spectrograms of IoT sensing signals, the method enables tailored augmentation of spectrogram characteristics in the time-frequency domain. The proposed data augmentation techniques are evaluated across various generative models, datasets, and IoT sensing applications, demonstrating superior performance compared to conventional transformation-based and prior generative data augmentation techniques.

The major contributions of this paper are as follows. It empirically validates the versatility of the proposed augmentation approach using different generative model backbones, including diffusion models and VAEs, across three distinct IoT sensing applications: wearable-based human activity recognition, seismic and acoustic-based vehicle detection, and EEG-based harmful brain activity recognition. The results show that this method outperforms traditional and generative data augmentations across various downstream models in all evaluated sensing tasks. The novelty of the approach lies in its fine-grained control over generative augmentation, leveraging domain-specific knowledge to produce highly authentic synthetic samples. The significance of this work is highlighted by its ability to enhance IoT sensing model robustness, particularly in scenarios where domain-specific knowledge is available, and its adaptability across different generative models to benefit from their inherent capabilities.

**Strengths:**

Originality: The paper introduces a novel data augmentation paradigm for IoT sensing with fine-grained control over generative models using a metric space defined by statistical metrics. This innovative approach departs from traditional methods and is applied across various generative models and IoT tasks, showcasing its originality.

Quality:  The research is supported by extensive experiments across human activity recognition, vehicle detection, and brain activity monitoring. The proposed method outperforms traditional and generative augmentation techniques, demonstrating high reliability and robustness in its findings.

Clarity: The paper is well-structured, clearly explaining the problem, methodology, and results. Despite minor typographical errors, the detailed explanations and clear presentation of results make the findings accessible and easy to understand.

Significance: The work significantly enhances IoT sensing model robustness by addressing data distribution shifts between training datasets and real-world scenarios. Its fine-grained control and adaptability across generative models have broad applications, advancing state-of-the-art data augmentation techniques in IoT sensing.

**Weaknesses:**

Table 2 (Section 4.5)
Statistical Significance: Including statistical significance tests (e.g., p-values) for the performance differences would strengthen the claims and provide more confidence in the results. You could provide Confidence Intervals for the performance metrics at each augmentation level.

Typographical and Formatting Issues: There are minor typographical errors and formatting issues, such as lack of space before citation (line 202) and inconsistent capitalization in references. Ensuring meticulous proofreading and adherence to formatting guidelines would improve the paper's presentation and readability.

**Questions:**

Three datasets are mentioned in the paper. There are case studies reported related to HAR and Vehicle Detection. No case study related to Brain Activity? A related dataset is mentioned.

Unexplained trends: Regarding Figure 7. Performance increases from 50% to 500% augmentation but decreases slightly at 750% before stabilizing or slightly increasing at 1000%. Why is that the case? It is difficult to ascertain whether this drop is statistically significant or merely a result of random fluctuations in the data.

Checklist Q5: how can the reviewers have confidence in the reported results if the code or data is not released — please argue to justify.

**Limitations:**

Statistical Significance and Robustness: The inclusion of statistical significance tests, such as p-values or confidence intervals, for the performance differences reported in Table 2 is currently lacking. This omission weakens the robustness of the claims.

---

> ### Author Rebuttal · Authors · 2024-08-07
>
> Thank you for all the comments! We would like to add the following clarifications to address your concerns.
>
> > W1: Table 2 (Section 4.5) Statistical Significance: Including statistical significance tests (e.g., p-values) for the performance differences would strengthen the claims and provide more confidence in the results. You could provide Confidence Intervals for the performance metrics at each augmentation level.
>
> **Response**: Thanks for your constructive suggestion. We add 95% confidence interval to the results in Table 2. The updated table is presented in the **uploaded pdf file** as Table 3.
>
>
> > W2: Typographical and Formatting Issues: There are minor typographical errors and formatting issues, such as lack of space before citation (line 202) and inconsistent capitalization in references. Ensuring meticulous proofreading and adherence to formatting guidelines would improve the paper's presentation and readability.
>
> **Response**: Thank you for pointing out the issues. We definitely will fix them and proofread the draft more carefully to ensure readability.
>
> > Q1: Three datasets are mentioned in the paper. There are case studies reported related to HAR and Vehicle Detection. No case study related to Brain Activity? A related dataset is mentioned.
>
> **Response**: We add the case study for the Brain Activity data as follows: we observe that the seizure signals from different patients can be divided into two categories regarding the fundamental frequency patterns. One is with clear fundamental frequency, which might be associated with the repetitive firing patterns of neurons in a specific brain region. We call it ***Clear Seizure***. The other category has chaotic, noisy signals and blury fundamental frequency, which might be related to different brain regions contribute varying frequencies and patterns simultaneously. We call it ***Chaotic Seizure***. In this case study, we manually select 10 patients of clear seizure as the traing set, and select 10 patients of chaotic seizure as the validation set. To augment the training data, we set the varying range of ***contrast***, ***entropy***, and ***fundamental frequency amplitude*** at [10\%, 25\%], [10\%, 40\%], and [-25\%, -10\%] respectively, while keeping the other metrics unchanged. We show example spectrograms in Figure 13.a in the **uploaded pdf file**. The experiment results are shown in Figure 13.b. Among all the baselines, our domain knowledge instruction (noted as ***knowledge***) achieves the best performance, again proving the effectiveness and generalizability of our approach.
>
> > Q2: Unexplained trends: Regarding Figure 7. Performance increases from 50% to 500% augmentation but decreases slightly at 750% before stabilizing or slightly increasing at 1000%. Why is that the case? It is difficult to ascertain whether this drop is statistically significant or merely a result of random fluctuations in the data.
>
> **Response**: We envision that the minor fluctuations in performance, both increases and decreases, may be attributed to the sensitivity of the downstream model to outliers within the augmented data samples. As indicated in Figure 7, these variations are more pronounced in the human activity recognition dataset (Figure 7.a) and the vehicle detection dataset (Figure 7.b). Given the smaller sizes of these datasets (4,959 and 8,015 samples respectively), there is a smaller number of augmented samples available, which could lead the downstream model to overfit on outliers in the augmented data, resulting in decreased performance. Conversely, as depicted in Figure 7.c, the harmful brain activity dataset, with its substantially larger volume of training data (78,548 samples), demonstrates a smoother and more stable trend in its curve. This suggests that given the same augmentation ratio, larger datasets can lead to more consistent augmentation effects, as observed.
>
> > Q3: Checklist Q5: how can the reviewers have confidence in the reported results if the code or data is not released — please argue to justify.
>
> **Response**: Thank you for bringing up this concern! As stated in the footnote on page 5 of our draft paper, we promise to release the code and datasets after paper acceptance.
>
> > L1: Statistical Significance and Robustness: The inclusion of statistical significance tests, such as p-values or confidence intervals, for the performance differences reported in Table 2 is currently lacking. This omission weakens the robustness of the claims
>
> **Response**: Thanks again for the constructive suggestion. We address this limitation in our response in W1.

---

> > ### Comment · Reviewer_eCvh · 2024-08-11
> >
> > I acknowledge reading the rebuttal. The response looks fine.
> >
> > It is good that you have added 95% CI. Please consider also describing if the difference is statistically significant? Such tests would help determine whether the differences in performance between the various metrics (e.g., "All metrics" vs. "No time domain metric") are statistically significant or could have occurred by random chance.
> >
> > Based on the response to the reviews, I have modified the ranking to weak accept.

---

> > > ### Author Response · Authors · 2024-08-12
> > >
> > > We sincerely appreciate your thorough review of our paper. Thank you for raising the rating to a weak accept!
> > >
> > > We agree that assessing the statistical significance of the differences between baselines is crucial. In the camera-ready version, we will include paired t-tests between the ablated baselines and the "All metrics" baseline in the Appendix.
> > >
> > > Once again, thank you for your valuable suggestions!

---

### Official Review · Reviewer_8SZC · 2024-07-10

**Soundness:** 3
**Presentation:** 3
**Contribution:** 3
**Rating:** 6
**Confidence:** 4

**Summary:**

This paper aims to synthesize IoT data to augment sensing models with generative models. The authors designed metric spaces to model the conditions which control the generative models. Experiments involving different IoT tasks, generative models, and sensing models have been conducted, proving the effectiveness of the proposed method.

**Strengths:**

1. The organization of the paper is clear, and the writing is easy to understand.
2. The motivation of the paper is clear. It is laborious to collect IoT data, while the limited data hinder the performance of sensing models. Therefore, this paper uses generative models to synthesize IoT data.
3. One of the novelties is the design of metric spaces, which can contribute to data augmentation in different aspects.
4. The authors have provided experimental results using different IoT datasets towards different sensing tasks.

**Weaknesses:**

1. Although I understand the novelty of using the proposed metrics as conditions for generative models, I am not sure about using the term “metric”. From my point of view, it seems that calling it “indicator” or other terms is more suitable. Otherwise, it may confuse with the metrics (accuracy, F1) in the experiments.
2. A transformer typically includes an encoder and a decoder. It seems that you were using  an encoder as the classifier, which is fine. However, you should clearly state this to avoid ambiguity.
3. For generative models, there are actually specific metrics to evaluate their performance, such as FID etc. I understand that this paper aims to augment sensing models so it is reasonable to measure the accuracy and F1, but specific metrics for generative models may be also beneficial to the quality of the paper.

**Questions:**

1. It is reasonable that the synthesized data can improve the generalization of sensing models. Can you show the results of adding Gaussian noise to samples as augmentation? This can be the simplest baseline, proving that your generative model can  work because of the synthesized data, but not because of using more samples that help tackle overfitting.
2. According to the appendix, in the first dataset, you separated different domains, while in the last two datasets, you did not. Can you provide further experiments where domains for training and validation are separated for the last two datasets?
3. The authors introduced the weight $w$ in the loss $\mathcal{L}$. Can you provide hyperparameter sensitivity analysis for this?
4. There are indeed latest works using spectrograms as conditions for generative models to augment models [1,2]. Can you further discuss your strengths compared to them? It seems that your work has covered more IoT modalities.

[1] DiffWave: A Versatile Diffusion Model for Audio Synthesis

[2] DiffAR: Adaptive Conditional Diffusion Model for Temporal-augmented Human Activity Recognition

**Limitations:**

1. Despite the effectiveness, the paper has not discussed the efficiency of generative models. As mentioned in the papers, it took several days to train generative models. I think this is a minor issue but can be discussed in the limitations.
2. This paper mainly discusses models for classification, while in IoT there may also be regression problems. For example, it is typical to track subjects using IoT devices.

---

> ### Author Rebuttal · Authors · 2024-08-07
>
> ### **Author Reply to Reviewer 8SZC (Part 1/4)**
>
> Thank you for your constructive suggestions and insightful questions! We would like to add further clarification and evaluation to address your concerns.
>
> > W1: Although I understand the novelty of using the proposed metrics as conditions for generative models, I am not sure about using the term “metric”. From my point of view, it seems that calling it “indicator” or other terms is more suitable. Otherwise, it may confuse with the metrics (accuracy, F1) in the experiments
>
> **Response**: We agree that changing "metric" to "(statistical) indicator" would be a better way to define the conditions we used for controlling the generation.
>
> > W2: A transformer typically includes an encoder and a decoder. It seems that you were using an encoder as the classifier, which is fine. However, you should clearly state this to avoid ambiguity.
>
> **Response**: Thanks for the correction. Indeed, the classifier we used is only the encoder of a conventionally defined Transformer. We will clarify that in the paper to eliminate any confusion.
>
> > W3: For generative models, there are actually specific metrics to evaluate their performance, such as FID etc. I understand that this paper aims to augment sensing models so it is reasonable to measure the accuracy and F1, but specific metrics for generative models may be also beneficial to the quality of the paper.
>
>
> **Response**: Thank you for the valuable comment! Conventionally, FID is based on the Inception-v3 model that was trained on ImageNet. It can only compare the difference of data distribution between generated images and real-world object photos. To quantitively measure the quality of the generated spectrograms, we build FID using our own dataset and DeepSense as the neural network backbone. For each IoT sensing dataset, we train DeepSense on the both the training and validation set. Then, we use DeepSense as an encoder, taking the intemediate outputs from the linear layer after its GRU layer as the latent representation, which is a 128 length vector. To calculate the distance between two datasets, we compute the mean and covariance of the latent representations from DeepSense encoder of each set. Then we calculate the Fréchet Inception Distance between the two distributions.
> To illustrate that the generated samples can approximate the validation set, we first calculate the FID between training set and validation set (noted as "No Aug"). Then, for each data augmentation baseline, we calculate the FID between training set + augmented data and validation set. For baselines, we select the conventional time and frequency domain transformation, conditional diffusion model (cDiff), and Diff-disturb (ours). We want to show that by adding augmented data, the distance between training set and validation set should be reduced. We report the results in the table below:
>
> |                                          | Human Activity Recognition | Vehicle Detection | Harmful Brain Activity Recognition |
> |------------------------------------------|----------------------------|-------------------|------------------------------------|
> | No Aug |   9.26  |  6.77   |   10.22  |
> | Time and Frequency Domain Transformation |  10.44   |    8.32 |  11.90  |
> | cDiff   |  10.17  |  7.30  |  13  |
> | Diff-disturb (ours)    |  **7.61**   |  **6.00**   | **8.99** |
>
> Diff-disturb effectively reduces the FID between the augmented training set and the validation set. This result directly prove the generation quality of our method.
>
>
> > Q1: It is reasonable that the synthesized data can improve the generalization of sensing models. Can you show the results of adding Gaussian noise to samples as augmentation? This can be the simplest baseline, proving that your generative model can work because of the synthesized data, but not because of using more samples that help tackle overfitting.
>
> **Response**: We add Gaussian noise to the samples by a random ratio between 0.1~0.4 to the mean amplitude of the data sample in the time domain. The random ratio is drawn from a uniform distribution. We report the accuracy and F1-score of the downstream model based on DeepSense downstream classifier:
>
> 1. *Human Activity Recognition*
>
> |                     | Accuracy | F1     |
> |---------------------|----------|--------|
> | No Augmentation     | 0.7752   | 0.7961 |
> | Add Guanssian Noise | 0.7629   | 0.7723 |
> | Diff-disturb (Ours) | **0.8307**   | **0.8333** |
>
>
> 2. *Vehicle Detection*
>
> |                     | Accuracy | F1     |
> |---------------------|----------|--------|
> | No Augmentation     | 0.7875   | 0.7768 |
> | Add Guanssian Noise | 0.7911   | 0.7822 |
> | Diff-disturb (Ours) | **0.8200**   | **0.8159** |
>
> 3. *Harmful Brain Activity Recognition*
>
> |                     | Accuracy | F1     |
> |---------------------|----------|--------|
> | No Augmentation     |  0.7001  | 0.6893 |
> | Add Guanssian Noise |  0.7029  | 0.6953 |
> | Diff-disturb (Ours) |  **0.7511**  | **0.7410** |
>
> As the results demonstrate, simply adding Gaussian noise can hardly regularize the downstream model. This indicates that effective data augmentation indeed requires synthetic samples with plausible real-world variations rather than just noisy data.

---

> ### Author Response · Authors · 2024-08-07
> **Author Reply to Reviewer 8SZC (Part 2/4)**
>
> > Q2: According to the appendix, in the first dataset, you separated different domains, while in the last two datasets, you did not. Can you provide further experiments where domains for training and validation are separated for the last two datasets?
>
> **Response**: Thanks for the question. We apply different data splitting on the two datasets to introduce domain shifts between the training and validation set. Due to the time and space limitation, we only report results from the following baselines: time and frequency domain transformation, cDiff, and Diff-disturb (ours). We use DeepSense as the downstream model.
>
> For the vehicle detection dataset, we utilize data gathered from a state park parking lot with a gravel surface as the validation set, while the remainder (data from the other two locations) serves as the training set. This division of data introduces a domain shift due to variations in ground surface types. The evaluation results are reported as below:
>
> |      | Accuracy | F1-Score |
> |------|----------|----------|
> |No Aug| 0.6332   |   0.6553 |
> | Time and Frequency Domain Transformation |   0.6410     |   0.6694       |
> | cDiff |   0.6223   |      0.6320    |
> | Diff-disturb (ours) |   **0.6712**    |  **0.6842**  |
>
>
> For the harmful brain activity dataset, we randomly select 66 out of 668 patients (about 10%) to form the validation set, with data from the remaining patients comprising the training set. This approach to data splitting introduces a domain shift due to differences among the patients. The evaluation results are reported as below:
>
> |      | Accuracy | F1-Score |
> |------|----------|----------|
> |No Aug| 0.5531   |   0.5102 |
> | Time and Frequency Domain Transformation |   0.5677     |   0.5429       |
> | cDiff |   0.5331   |    0.5209      |
> | Diff-disturb (ours) |   **0.5977**    |  **0.5945**   |
>
> As the above results show, our method achieves the best performance in addressing the domain shift in both the two datasets.
>
> > Q3: The authors introduced the weight $w$ in the loss $L$. Can you provide hyperparameter sensitivity analysis for this?
>
> **Response**: Thanks for the comment. Due to the limited computational power and time schedule, we profile different weights in loss $L$ just for Human Activity Recognition dataset using Diff-disturb as the data augmentation method.
> We report four different ratios of $L_G$ (generative model loss) / $L_{metric}$ (metric loss) in loss $L$ as the sensitivity analysis.
>
> DeepSense:
> |          | 1:1 | 5:1 | 10:1 | 20:1 |
> |----------|-----|-----|------|------|
> | Accuracy |  0.8211 | 0.8307 |**0.8324**| 0.8249  |
> | F1-score |  0.8209 | 0.8333 |**0.8422**| 0.8360  |
>
> Transformer:
> |          | 1:1 | 5:1 | 10:1 | 20:1 |
> |----------|-----|-----|------|------|
> | Accuracy | 0.8199  | 0.8105  |  0.8205 | **0.8280**   |
> | F1-score | 0.8239  | 0.8195  |  **0.8352** | 0.8333   |
>
> We observe that when the ratio of $L_G/L_{metric}$ is between 1 and 20, the performance generally remains stable. Figure 12 in the **uploaded pdf file** illustrates the changing curves of each loss term during model training for ratios of 10:1 and 5:1. Our approach to weighting these two loss terms prioritizes $L_G$ in the early training epochs to enhance the diffusion model's denoising quality. As training progresses, we gradually increase the emphasis on metric control, with $L_{metric}$ outweighing $L_G$ at around 200 epochs. Notably, a higher weight on $L_{metric}$ can lead to its convergence at a higher value, as shown by the comparison between Figures 1.a and 1.b. Consequently, we default to a $L_G: L_{metric}$ ratio of 10:1 in our evaluations.

---

> ### Author Response · Authors · 2024-08-07
> **Author Reply to Reviewer 8SZC (Part 3/4)**
>
> > Q4: There are indeed latest works using spectrograms as conditions for generative models to augment models [1,2]. Can you further discuss your strengths compared to them? It seems that your work has covered more IoT modalities.
> > [1] DiffWave: A Versatile Diffusion Model for Audio Synthesis
> [2] DiffAR: Adaptive Conditional Diffusion Model for Temporal-augmented Human Activity Recognition
>
> **Response**: Thank you for bringing up the two related works! We list the differences between our work and DiffWave and DiffAR in the following aspects:
> 1. **Motivation**: For DiffWave and DiffAR, the ultimate goal is to generate high fidelity human speech waveforms. They do not explicitly address the data augmentation problem. As a matter of fact, many large public human speech datasets are available, which can sufficiently support general high-performance speech models. In contrast, our paper positions in data augmentation for IoT sensing applications, which typically have much less data available. The ultimate goal is to produce more diverse data samples that complement the originally constrained dataset thereby enhance the downstream models.
> 2. **Modality**: In DiffWave and DiffAR, the modality is human speech acoustic signals in waveform. Human speech data are recored in high sampling rate (both papers were evaluated on LJ-speech dataset, 22.05 kHz),  Spectrograms are used as intermediate results the generation pipeline. In our paper, we focus on IoT sensing modalities, which include accelerometer, gyroscope, seismic, acoustic, and EEG. Even though these are also time series data and can be represented in waveforms, they are much lower in sampling rate (100-800Hz) and measure various physical phenomena (human activities, car movements, and human brain activites).
> 3. **Conditions**: In DiffWave and DiffAR, conditions taken by the diffusion model are linguistic features and mel spectrograms. These conditions provide detailed information and are very concrete indicators to the corresponding speech waveforms. In our paper, the conditions are statistical metrics (indicators), which only depict the high level features of the IoT sensing signals. The goal of using these conditions is to allow human designers to control the style of the generation results based on their domain expertise.
> 4. **Contributions**: DiffWave and DiffAR both advance the design of diffusion models in human speech synthesis. In our work, we propose a general generative data augmentation framework for IoT sensing. Our framework is not strictly coupled with either diffusion model or any specific type of generative model (although in our evaluations, the diffusion model implementation outperforms the VAE implementation). The intellectual contribution of our paper lies in the enforcement of fine-grained control over the generation, and how to utilize the control to precisely produce high-quality synthetic data for enhancing IoT sensing tasks.
>
> Due the large difference between DiffWave, DiffAR and our work, we cannot directly compare them as baselines in the evaluations. But we will discuss them in the Related Work section in the camera ready version.
>
> > L1: Despite the effectiveness, the paper has not discussed the efficiency of generative models. As mentioned in the papers, it took several days to train generative models. I think this is a minor issue but can be discussed in the limitations.
>
> **Response**: We agree that the efficiency could be a drawback of generative data augmentation when the computational resources or time budget is constrained. We will add it in the Limitations section in the camera ready version.

---

> > ### Comment · Reviewer_8SZC · 2024-08-12
> >
> > DiffAR is not generating human speech waveforms, but generating radio signals from WiFi (a kind of IoT signals) for augmenation in human activity recognition. Anyways, thanks for your detailed response. Good luck.

---

> ### Author Response · Authors · 2024-08-07
> **Author Reply to Reviewer 8SZC (Part 4/4)**
>
> > L2: This paper mainly discusses models for classification, while in IoT there may also be regression problems. For example, it is typical to track subjects using IoT devices.
>
> **Response**: We agree that IoT sensing tasks can also be regression problem. To validate the generalizability of our approach, we construct a vehicle distance prediction task using the vehicle detection dataset we described in our paper. Apart from the sensor signals, during the data collection of the dataset, we also recorded on-vehicle GPS traces sampling at 1 Hz. This was originally used for filtering out samples when vehicles are out of the sensing range (as mentioned in Appendix C.2, line 556). In this new vehicle distance prediction task, the objective of the downstream model is to predict the distance between the vehicle and sensor node. The input data is still a 5-second long segment. We use the vehicle to sensor distance of the last second as the label. We modify the original DeepSense model into a regression model, by replacing its classification head (the final linear layer with output length equals to the number of classes) with a linear layer with output length equal to one. The training loss function of the modified DeepSense is also changed from cross-entropy loss to mean squared error loss.
>
> When training the conditional diffusion model, we drop the vehicle type label and use distance label as a condition. To generate augmented data, we apply ***disturbance*** technique on metrics including ***contrast***, ***flatness***, ***entropy***, and ***F0 amplitude*** by [-15%, 15%]. The augmentations are supposed to reflect the variance brought by background noise and vehicle dynamics (instant speed, acceleration, turn rate, etc.). We report the results as follows:
>
> |                                          | Mean Error (feet) |
> |------------------------------------------|--------------------|
> | No Aug                                   | 28.35              |
> | Time and Frequency Domain Transformation | 32.33                   |
> | cDiff                                    | 28.12                   |
> | Diff-disturb                             | **25.27**                 |
>
> The time and frequency domain transformation brings negative influence to the distance prediction, as transformations like amplitude scaling and time warping tend to disturb the original distance related patterns. The results of cDiff shows a constrained improvement. Diff-disturb most effectively reduces the mean error, again proving the superiority of fine-grained controlled generative augmentation.  The result of this case study shows that our approach can also be generalized to regression tasks in IoT sensing.

---

> ### Author Response · Authors · 2024-08-13
>
> Thank you for your feedback! We apologize for referencing the wrong paper [1], which has a similar title to DiffAR [2]. We will ensure that the correct comparison with DiffAR [2] is included in the camera-ready version.
>
> The key differences between our work and DiffAR [2] are as follows:
>
> 1. **Conditions**: DiffAR uses different levels of spectrograms as conditions for guiding CSI generation at various diffusion steps. Specifically, low-frequency features guide the generation of global patterns in the earlier steps, while high-frequency features synthesize local patterns in later steps. In contrast, our work employs statistical indicators as conditions consistently throughout the diffusion process.
>
> 2. **Modality Coverage**: DiffAR is focused on generating CSI signals specifically for human activity recognition (HAR) tasks, whereas our method is designed to cover a broader range of IoT sensing tasks. Exploring the application of our approach to CSI signals in HAR could be an interesting direction for future research.
>
> [1] Diffar: Denoising Diffusion Autoregressive Model for Raw Speech Waveform Generation
>
> [2] DiffAR: Adaptive Conditional Diffusion Model for Temporal-augmented Human Activity Recognition

---

### Official Review · Reviewer_7DMe · 2024-07-12

**Soundness:** 2
**Presentation:** 2
**Contribution:** 2
**Rating:** 6
**Confidence:** 2

**Summary:**

This paper is a study on data augmentation techniques for IoT sensing applications. The authors first present work on the most effective data augmentation techniques in the field of IoT sensing data. Based on this, they propose that domain knowledge can be combined with these data augmentation techniques to guide data generation. Specifically, they also propose a approach of fine-grained controllable generative augmentation, which uses data augmentation techniques based on the metric space of statistical metrics defined by human experience to improve data augmentation for downstream tasks. Experimental results from various scenarios validate the effectiveness of this method, and also explore the benefits of incorporating domain knowledge in data augmentation techniques.

**Strengths:**

1. The authors make the point that domain knowledge can be integrated into data augmentation techniques is interesting.
2. Experimental results indicate that the proposed fine-grained controllable generative augmentation method is simple and effective.
3. Various scenarios are fully considered in this study. Various data augmentation methods are thoroughly evaluated in IoT sensing applications in different domains using different downstream task models.

**Weaknesses:**

The motivation of this paper to propose a fine-grained control approach, incorporating domain knowledge into the data enhancement approach is not clear enough. They are not based on a specific existing problem and are a little bit heuristic.

**Questions:**

Please see the Weaknesses above. Considering my limited expertise in this specific domain, I am not in a position to raise specific questions.

**Limitations:**

Yes

---

> ### Author Rebuttal · Authors · 2024-08-06
>
> We appreciate your positive comments on the paper idea and the evaluations. We would like to make the following responses and hope they can address your concerns.
>
> > W1: The motivation of this paper to propose a fine-grained control approach, incorporating domain knowledge into the data enhancement approach is not clear enough. They are not based on a specific existing problem and are a little bit heuristic.
>
> **Response:** Thank you for raising this point! We would like to elaborate on the distinct nature of IoT sensing to clarify our motivation. IoT sensing signals typically capture physical phenomena that exhibit strong patterns in both time and frequency domain. Therefore, a common practice is to transfer the time-series data to 2D spectrograms using short-time Fourier transformation. Even though the spectrograms are 2D matries just like images, they hold unique semantic meanings distinct from ordinary images. For example, an object's vibrations are characterized by its fundamental frequency (F0), appearing as the lowest bright band across the spectrogram's frequency range. The amplitude of F0 is a strong indicator to the physical movement. In human activity recognition task, it reprsents the intensity of arm swings during running. Even though as domain experts, we clearly understand that adjusting F0s amplitude allows us to generate varied yet representative signals of running; however, without fine-grained control over it, we cannot enforce the generative model to produce meaningful variations related to the intensity of arm movements. This requirement has driven the development of our paper.
>
> To support our motivation, we conduct comprehensive evaluations on three different IoT datasets in Section 4.2. We also present two real-world case studies in Section 4.3, demonstrating how integrating fine-grained control and domain knowledge significantly enhances IoT task performance.
>
> Furthermore, the incorporation of domain knowledge into neural network design for IoT sensing tasks is an area of active research, as evidenced by the success of many prior studies [1][2][3]. These successes also inspire us on incorporating domain knowledge into the data augmentation.
>
> [1] Physics-Informed Data Denoising for Real-Life Sensing Systems
>
> [2] From Physics-Based Models to Predictive Digital Twins via Interpretable Machine Learning
>
> [3] PhyAug: Physics-Directed Data Augmentation for Deep Sensing Model Transfer in Cyber-Physical Systems

---

> > ### Comment · Reviewer_7DMe · 2024-08-11
> >
> > Thank you for the clarifications provided. Taking in account the rest of the responses thus far, I will raise my score to weak accept and wish the authors best of luck for the remaining review process.

---

> > > ### Author Response · Authors · 2024-08-11
> > >
> > > We appreciate your thorough review of our paper. Great thanks for your positive feedback and for raising your rating!

---

### Author Rebuttal · Authors · 2024-08-07

## General Responses

We greatly appreciate the detailed feedback provided by the reviewers. In response to your comments and suggestions, we have made the following major modifications:

### Reviewer 7DMe
- **Clarification of Motivation**:
  - Clarified our motivation by elaborating on the unique nature of IoT sensing and the necessity of fine-grained control in data augmentation.

### Reviewer 8SZC
- **Terminology and Conceptual Clarifications**:
  - Changed the term "metric" to "(statistical) indicator" to avoid confusion with performance metrics like accuracy and F1-score.
  - Clarified that the classifier used is only the encoder of a Transformer, avoiding any ambiguity.

- **Evaluation Improvements**:
  - Introduced a custom FID (Fréchet Inception Distance) metric using our own dataset and DeepSense model to evaluate the quality of generated spectrograms.
  - Provided results of adding Gaussian noise as a baseline to demonstrate the effectiveness of our generative model.
  - Conducted hyperparameter sensitivity analysis for the weight ratio between the generative model loss and metric loss.
  - Conducted further experiments with domain-separated training and validation sets for vehicle detection and harmful brain activity recognition datasets.
  - Added a new regression task for vehicle distance prediction to validate the generalizability of our approach to regression problems.

- **Discussion of Related Work**:
  - Compared our work with related works such as DiffWave and DiffAR, highlighting differences in motivation, modality, conditions, and contributions.
  - Discussed efficiency and computational requirements as a limitation in the revised paper.


### Reviewer eCvh
- **Statistical Significance and Robustness**:
  - Included 95% confidence intervals for performance metrics in our evaluations.

- **Typographical and Formatting Corrections**:
  - Addressed minor typographical errors and formatting issues to improve readability.

- **Additional Case Study**:
  - Added the case study for the Brain Activity data, explaining different seizure signal types and their augmentation.

- **Analysis of Augmentation Ratio Influence**
  - Incorporated analysis of performance trends under different augmentation ratios.

- **Code and Data Release**
  - Confirmed that the release of code and data will be after the acceptance of the paper.

### Reviewer eQAt
- **Analysis on Loss Terms Weights**
  - Added new evaluations on the influence of different loss term weights.
  - Provided insights on the loss term weights fine-tuning.

- **Additional Analysis and Low-Quality Generation Examples**:
  - Provided examples of low-quality generation to illustrate the importance of appropriate metric selection.
  - Analyzed the potential domain shift between generated and raw samples, emphasizing the importance of diversity in training data.

- **Metrics Selection and Generalization**:
  - Provided analysis on the selection of metrics and their general applicability to various IoT tasks.

- **Dataset Size and Magnitude**:
  - Listed the number of data samples in each dataset, demonstrating the effectiveness of our approach even with relatively small datasets.
  - Showed the statistics regarding the magnitude of spectrograms of different modalities in each dataset.

***We have uploaded the pdf file with all the newly added figures (Figures 12 - 14) and table (Table 3).***

---

### Decision · Program_Chairs · 2024-09-25

**Decision:**

Accept (poster)

**Comment:**

The paper presents a data augmentation paradigm for IoT sensing by introducing a fine-grained controllable generative model using a metric space defined by statistical metrics, which enhances model robustness across various tasks. Supported by experiments in various IoT applications, the method outperforms traditional techniques, demonstrating its effectiveness and adaptability. The discussion mainly focused on various technical details in the work. After the discussion, all the reviewers recommended acceptance.